# Chemical and Mechanical Characterization of the Alternative Kriab-Mirror Tesserae for Restoration of 18th to 19th-Century Mosaics (Thailand)

**DOI:** 10.3390/ma16093321

**Published:** 2023-04-23

**Authors:** Thawatchai Ounjaijom, Pratthana Intawin, Arnon Kraipok, Surapong Panyata, Rachata Chanchiaw, Yunee Teeranun, Prapun Gaewviset, Pathoo Boonprakong, Ekarat Meechoowas, Terd Disayathanoowat, Samart Intaja, Phatcharaphon Dito, Choktavee Piboon, Kamonpan Pengpat

**Affiliations:** 1Department of Mechanical Engineering, Faculty of Engineering, Rajamangala University of Technology Lanna, Chiang Mai 50300, Thailand; tounjaijom@gmail.com; 2Division of Physics, Faculty of Science and Technology, Rajamangala University of Technology Thanyaburi, Pathum Thani 12110, Thailand; pratthana_i@rmutt.ac.th; 3Department of Physics and Materials Science, Faculty of Science, Chiang Mai University, Chiang Mai 50200, Thailand; kraipok.a@gmail.com; 4Faculty of Industrial Technology, Rambhai Barni Rajabhat University, Chanthaburi 22000, Thailand; surapong.p@rbru.ac.th; 5Rachata Chanchiaw’s Home Manufacture, 235, M. 2, T. Nongkaew, A. Hang Dong, Chiang Mai 50230, Thailand; 6Department of Fine Arts, The Office of Traditional Arts, Ministry of Culture, Phutthamonthon, Nakhon Pathom 73170, Thailand; 7Division of Engineering Materials, Department of Science Service, Bangkok 10400, Thailand; 8Department of Biology, Faculty of Science, Chiang Mai University, Chiang Mai 50200, Thailand; 9Samart Handcrafted Thin-glass Workshop, 48, M. 1, T. Nongkaew, A. Hang Dong, Chiang Mai 50230, Thailand

**Keywords:** Kriab-mirror tesserae, lead-coated mosaic mirrors, shear strength, adhesive bond, color parameters, accelerated weathering, restoration

## Abstract

Kriab-mirror tesserae are a type of lead-coated mosaic mirror found in most archaeological sites and antiquities dating back to the 18th century in central Thailand. The need for restoration work has prompted the search for alternative mirrors with similar characteristics to the ancient ones. Prototypes of alternative lead-coated mirrors were successfully used to restore a variety of archaeological sites and artifacts, demonstrating their potential application in heritage conservation and restoration work. We investigated the selected ancient Kriab samples in terms of their composition in both glass and reflective coating layers, as well as the chemical and mechanical characterization of the selected alternative Kriab mirrors. We employed a standard lab-shear test, which proved difficult to evaluate due to failure not occurring between the glass-to-metal interfaces. However, a modified lab-shear specimen setup was used to elucidate the peel-off bonding behavior of the lead-to-glass interface. Additionally, we measured the L*, a*, and b* values in the CIE-Lab standard, which exhibited variations for each colored Kriab mirror. The %reflectance of the selected ancient and alternative Kriab mirrors was highly similar when lower than a high %reflectance of a standard silvering mirror. Thai professional conservators have embraced the use of alternative Kriab mirrors in restoration projects as a replacement for old Kriab mirrors, as they are more compatible in terms of color and avoid the excessive brightness of silvered colored mirrors. However, the weathering durability of the alternative mirrors was poor due to the leaching of alkaline and lead ions caused by hydrolytic attack on the poor chemical stability separated phase. Overall, our research provides valuable insights into the properties and qualities of both ancient and alternative Kriab mirrors, which will be useful in the further development of mirrors with more resembling properties or even more environmentally friendly Kriab mirrors and their potential applications in restoration work in Thailand and archaeological sites in Asia.

## 1. Introduction

The use of glasses and mirrors for decorative purposes has a long history dating back to ancient times. Artifacts demonstrating such usage have been found worldwide, including in Egypt, Africa, Europe, and Asia. One particularly intriguing example of decorative mirrors is the lead-coated mirror. In a 2002 publication, Kock and Sode [1] reported on the discovery of medieval glass mirrors that may have been made by applying lead backing to a blown glass globe while still hot. They also noted that Kapadvanj Gujarat, India, might be the only remaining location in the world (as of 2002) producing lead-coated glass mirrors. These mirrors were used in a variety of applications, including embroidered textiles known as shisha, household utensils, furniture, small figures, and wall decorations. Interestingly, in the 17th century, the Mughals and others used these mirrors in their palaces. The spread of the mirror-making process and products into Thailand might be attributed to the long-term relationship between India and Thailand. Based on the findings of our survey, it was revealed that lead-coated glass-mosaic mirrors were present in neighboring Thai countries, including temples located in Laos and Myanmar. However, to ascertain the veracity of this observation, non-destructive techniques may be employed, and an inquiry into examining ancient samples would necessitate authorization.

Kriab-mirror tesserae, which are lead-coated mirrors, have been discovered in various archaeological sites and antiquities in the central region of Thailand. These mirrors were used for ornamental purposes, particularly in royal palaces and temples. Previous studies [2,3,4,5,6] have revealed that Kriab mirrors were made of colored glasses with a reflective coating of lead-based alloys and that they had a glass thickness of 0.3–0.7 mm and a thin coating layer of less than 0.05 mm. However, the process of manufacturing these mirrors remains unknown. The only available source of information is an antique book titled “Tumra Hung Krachok Kriab (Production of Kriab mirrors)”, discovered in the 1860s by His Royal Highness Prince Krom Khun Worachak Phalanubhab, which contains raw material and wage information, and the names of workers written in archaic Thai, without any explanation of the production process [7]. It is challenging to determine whether the manufacturing process of these Kriab mirrors was transferred from India. However, researching the production of Kriab mirrors is crucial, as it can provide insights into the manufacturing techniques used in ancient times and contribute to the preservation and understanding of these valuable artifacts. Moreover, this research can help artisans avoid using commercially available colored mirrors coated with silver, which are too bright and look alienated in restoration work.

In recent years, our team has successfully fabricated alternative Kriab mirrors using modified tape casting techniques, which were originally invented by local Chiang craftsmen led by co-author Chanchiaw. These mirrors have been widely distributed for renovation and conservation purposes. Notably, specialists from the Office of Traditional Arts, Department of Fine Arts, and Ministry of Culture in Thailand have utilized them in the renovation projects of historical artifacts such as the Narai on Suban Royal Yacht, Royal Vehicles, and Yanam during the reign of King Rama IX.

According to Klysubun et al. [5], some Kriab mirrors obtained from the Temple of the Emerald Buddha consist of soda-lime silicate with a PbO content ranging from 2 to 16 wt% for yellow and colorless mirrors. However, the same group of researchers found that red-colored Kriab samples contained more PbO (46–52 wt%) [6]. In a separate study by Wong-In and his coworkers [2], the chemical compositions of fourteen ancient Thai mirrors collected from the same temple were analyzed and found to contain 20–50 wt% PbO, depending on the glass colors, with red-colored glasses having the highest amount of PbO, consistent with the findings in [6].

To understand why ancient Thai artisans preferred lead-based glass, it is important to look at the historical context of lead glass production. Records indicate that lead has been used in glasses for thousands of years, with evidence of its use as early as 1400 BC [8,9,10,11,12]. A study of ancient glass beads from the 1st and 2nd centuries BC in South and Southeast Asia, including Korea, revealed that some samples contained high-lead glasses, which were more common in Korea and might have been made in China [11]. In the 2nd century AD, Roman glassmakers used high concentrations of lead and copper to produce red opaque glass mosaics [12]. During the medieval period in Central Europe, various types of lead vessel glass were manufactured, including soda-lead glass, wood-ash lead glass, and high-lead glass, as reported in [10]. While there is debate over who first used lead in glass, evidence suggests that European missionaries introduced glass enameling techniques to Asian craftsmen, first in Japan and then in China, at the end of the 17th century [13]. Although lead glass was produced in smaller quantities compared to classical natron and wood ash glasses, it was necessary to add PbO to the glass batch to lower the melting temperature to a range of 900–1000 °C, significantly lower than that required for wood ash or natron glasses, which exceeded 1200 °C [9,10,14].

The Silk Road, which facilitated the exchange of scientific, technological, artistic, and religious ideas among different civilizations and peoples in the region, may have played a significant role in the dissemination of knowledge about lead glass. It is possible that Thai artisans learned about the benefits of using lead glass through interactions with European and/or Chinese traders and may have even discovered their own color glass recipes. Hence, the use of PbO by Thai artisans to lower the melting temperature of the glass is understandable.

Chanchiaw and his colleagues developed a glass recipe that incorporated a high concentration of PbO (>50 wt%) in sand and borax. This was necessary to ensure that their initial electric furnace, which had a maximum temperature of approximately 900 °C, could be utilized effectively. However, due to the handmade production process, maintaining consistent quality was challenging, as each glass batch varied in characteristics such as color and adhesive strength to the lead-based alloy used for reflecting metal. The use of lead as a reflective coating in our alternative Kriab mirrors presents a significant advantage over commercially available silver-colored mirrors. The latter can appear too bright when juxtaposed with ancient mirrors.

This study aims to investigate the ancient and early prototype alternative Kriab-mirror tesserae, with a particular emphasis on their glass-to-metal interfaces. Unlike previous studies [2,3,4,5,6], this research examined Kriab mirror samples from various artifacts. X-ray diffraction (XRD) analysis was employed to determine the phase composition of the reflective coating of the ancient Kriab samples while scanning electron microscopy (SEM) was used to study the interface between glass and metal layers and their chemical composition. Additionally, the optical properties, such as color and reflectivity, of the alternative Kirab mirror samples, the first prototypes of lead-coated mirrors, were investigated. The ancient Kriab-mirrors were often used as outdoor decoration for the walls of temples and palaces, as well as for some outdoor statues. The alternative Kriab-mirror samples were carefully examined for their suitability in harsh conditions, such as UV exposure, high temperatures, and humidity in Thailand, in terms of weathering. To understand the nature of bonding between the glass surface and the lead coating, the adhesive property of the glass and lead-coated layer of the alternative Kriab samples was evaluated by Single-lap-Joint Testing. This may aid in the future development of coating techniques, not just for lead but also for non-lead metal alloys, in more environmentally friendly alternative prototypes.

## 2. Materials and Methods

### 2.1. Investigation of the Ancient Kriab Mirrors by X-ray Diffraction and Scanning Electron Microscopy

Rather than the more traditional gold-on-red lacquer painting [15,16], ancient Kriab mirrors were decorated with mosaic art, as shown in Figure 1. Depending on the imaginative design, mirror-cutting pieces could vary in size from 0.5 to 1.5 cm in length and come in a variety of shapes (Figure 2). The reflective coating of the ancient Kriab mirrors was analyzed using X-ray diffraction (PANalytical, X’Pert-Pro MPD, Almelo, Netherlands) to collect data in a 2θ range of 10° to 60° at a step size of 0.012°, using Ni-filtered CuK_α_ radiation at 60 kV and 80 mA. To determine the precise composition of the glass and reflective coating metal, a scanning electron microscope (SEM, JSM-IT300LV, Japan Electron Optics Laboratory Co., Ltd. (JEOL), Tokyo, Japan) operating at a voltage of 15 kV and equipped with a Wavelength-dispersive X-ray spectroscope (WDS) detector was utilized. At least three points of WDS analysis were determined in each selected ancient Kriab sample. It can be noted that with the proper standard, the light element Boron can be detected. A cross-sectional analysis showing X-ray mapping of selected ancient Kriab mirrors was carried out using a scanning electron microscope (FE-SEM, SU5000, Hitachi High-Tech Corporation. Hitachi, Tokyo, Japan) equipped with an Energy-dispersive X-ray spectroscope (EDS) detector.

### 2.2. Production of Alternative Kriab Mirrors

As mentioned in the introduction, the first available furnace had a maximum temperature of 900 °C, which led to the initial glass batch for Kriab mirror production containing a high lead content, such as a glaze in ceramic production with an approximate composition of 6.4% SiO_2_, 86.4% PbO, and 7.2% Na_2_[B_4_O_5_(OH)_4_]·8H_2_O (borax) by weight. After collaborating with our research team and obtaining a higher-temperature electric furnace with a maximum temperature of 1200 °C, we continued to use the same base glass. This was because the modified tape casting approach required a low viscosity melt for the fluid glass to flow through the narrow slit and form a thin glass sheet on top of a warmed steel plate (250–300 °C) to avoid sudden temperature changes. The resulting thin glass sheet has an area of approximately 20 × 80 cm^2^ and varies in thickness between 0.5–0.7 mm. Then, another layer of lead melt was coated on top of the hot glass sheet to create the Kriab mirror, which was cooled to room temperature for production. Commercial pure lead metal was used in the first prototypes. Approximately 3 kg of the batch was melted each day, and the colorant oxide was added empirically until the color of the test filament was determined visually to be correct by the workers before forming the thin sheet. The residual glass was remelted, allowing for the production of approximately 15 to 25 sheets of Kriab mirrors per day. Every worker was concerned about the toxicity of lead and took precautions to protect themselves. The modified tape-casting technique was initially invented by Chanchiaw and his coworkers to avoid the classical mouth-blowing technique, which was riskier due to the potential exposure to lead vapor.

To confirm the composition of our alternative Kriab mirror products, we performed SEM-EDS (Scanning Electron Microscopy—Energy Dispersive X-ray Spectroscopy) using the S-3400N instrument from Hitachi, Tokyo, Japan, operating at 15 kV. This allowed us to determine the chemical compositions of the selected colored Kriab mirrors on the glass and lead layers. The newly employed SEM-EDS is capable of detecting light elements such as boron. Table 1 presents the findings, which encompass clear, blue, green, and red glass layers. For the lead layer, this EDS method determined about 80 wt% and a small amount of 16 wt% C and 4 wt% O, and no other impurity was found with the limitation of EDS analysis. In the glass layers, as shown in Table 1, the EDS analysis mentioned above could not detect the small quantity of colorant in the glass layers, such as CoO for blue glass, the presence of the element B was also discernible. When the glasses were melted at high temperatures, the molten material corroded the crucible surface, resulting in a minor quantity of alumina (Al_2_O_3_). With the exception of red glass, the clear, cobalt blue, and emerald green glass samples were found to contain high levels of CuO (2–3 wt%). It can be noted that green glass is intentionally produced by adding 3 wt% CuO to the glass composition. Furthermore, different EDS analyses with SEM (JSM-IT800, Japan Electron Optics Laboratory Co., Ltd. (JEOL), Tokyo, Japan) at 15 kV detected trace amounts of 0.02 weight percent CuO and 0.09 weight percent SnO_2_, which were shown to contribute to the red chromophore in red glass. The presence of CuO impurities in the clear and blue glass samples may be attributed to the use of the same crucible for melting multiple-colored glasses in our glass workshop, which generally lasts between 5 and 7 days. It is noted that the red Kriab mirror was melted separately from the other colored Kriab mirrors.

### 2.3. Single-Lap-Joint Testing for Adhesive Property of the Alternative Kriab Mirrors

Numerous alternative methods are available for assessing the glass-to-metal bond. However, this study employed the single-lap-join under static tensile test, following the ASTM D1002-99 standard [17], which is simple and suitable for delicate mirror samples. Figure 3 shows the schematic diagrams of the ASTM standard for the single-lap-joint specimen, which utilizes two clamping steel substrates made from 1.5 mm S275 steel plates. Tensile tests were conducted using a Hounsfield H50KS universal test machine (UTM), with a constant crosshead speed of 1 mm/min, in accordance with ASTM D1002-99. The 625 mm^2^ sample area was bonded to the steel substrates using the same adhesive used for the overlap. The alternative Kriab mirror samples were investigated in four distinct colors, with at least three specimens tested for each glass color investigation. The average result of each specimen was used for the analyses, and the shear strength was calculated using Equation (1) [18].
(1)$$\tau =\frac{P}{A}$$
where τ is tensile shear strength (N/m^2^), P denotes breaking force (N), and A is a shear area.

The study utilized the Scotch-WeldTM Epoxy Adhesive DP100 Plus Clear, a high-strength two-part resin, to bond the glass samples to the two clamping steel plates. To evaluate the shear strength of the epoxy resin sample, a standard test of three specimens was performed using the same configuration as depicted in Figure 3. Figure 4 illustrates the experimental setup of the ASTM standard single-lap-joint under the static tensile test method of the epoxy adhesive. The manufacturer data for the epoxy resin used in this test are listed in Table 2. This epoxy-based adhesive is specifically designed to bond glass and metal, and it possesses high crash resistance. Prior to bonding the metal substrates with the epoxy adhesive, all glass and lead-based alloy surfaces were scratched with a high-hardness metal tool, as shown in Figure 5. The metal substrates’ surfaces were polished with sandpaper 180 and degreased with isopropanol. This surface preparation was conducted to ensure the best possible adhesive bond between the epoxy adhesive and both specimen surfaces.

### 2.4. CIE-Lab Color Measurement and Reflectivity

To measure the color of the ancient Kriab and prototype mirrors, we used a Konica Minolta CM-3500D spectrophotometer (Konica Minolta, Osaka, Japan) at room temperature, with a daylight CIE-D55 illuminant and a diffuse/illuminating/measuring geometry. We measured the color in the CIE-Lab* color space. To ensure accuracy, we calibrated the equipment using a standard CR-A43 white calibration plate (with a*/0.3156, b*/0.3319, L*/93.80). We used the standard silvering mirror as a reference to measure %reflectance.

### 2.5. QUV Accelerated Weathering Test

A controlled accelerated weathering test was used to assess the weathering resistance of the alternative Kriab mirrors. A QUV machine (QUV Accelerated Weathering type QUV/SE) was used to test a variety of Kriab mirrors in various colors. For a period of 14 days (336 h), alternate cycles of UVB 313 nm radiation (8 h) at 60 °C and 0.63 W/m^2^/nm irradiance and water condensation (4 h) at 50 °C were conducted according to the ASTM G 154-00a standard (a practice for operating fluorescent light Apparatus for UV exposure of non-metallic materials). The effects of the accelerated weathering tests on the alternative Kriab specimens were evaluated in terms of visual observations and color variation. Reflectance spectra of the selected mirrors before and after a QUV for 14 days were measured using a standard silvering mirror for calibration. After the test, the surface of the selected Kriab mirrors was examined by a JEOL JSM-IT300LV scanning electron microscope (SEM).

## 3. Results and Discussion

### 3.1. Phase, Chemical Composition, and Cross-Sectional Analysis of the Ancient Kriab-Mirror Tesserae

We obtained and analyzed samples during the restoration of historic Kriab mirrors. It should be noted that the back of these ancient Kriab mirrors contains a reflective coating layer and a natural glue called Rak Samuk. The Rak Samuk technique, a lacquering method involving boiled resin mixed with burnt powder from banana leaves, bricks, or coconut shells, was commonly used to prepare the surfaces of wooden handicrafts and architectural surfaces. Evidence suggests that this technique originated in China [20]. Figure 6 shows the XRD patterns of the coatings on some of the Kriab mirrors that make them reflect light. Lead metal was found to be the significant phase of most Kriab samples. Some samples, such as Kriab 1, showed only a lead phase. Tin was found in small quantities in some Kriab examples, such as Kriab 2 and Kriab 3. Most of these samples have degraded over time and fallen out.

Lead oxide (PbO), basic lead carbonate (Pb_2_CO_3_Pb(OH)_2_), commonly known as hydrocerussite, and sodium lead carbonate hydroxide (NaPb_2_(CO_3_)_2_(OH)_2_) were identified as secondary phases in some of the Kriab samples. Hydrocerussite was found in most of the Kriab samples, including Kriab 3. Sodium lead carbonate hydroxide was only detected in Kriab 4 sample. The reflective coating on samples with pure lead (Kriab1) and lead/tin (Kriab2) coatings showed preferred orientations in the (220) and (200) planes, respectively. Heterogeneous nucleation may have occurred during the casting of these metal alloys. The difference in composition between the two coatings played an important role in the solidification of this Pb–Sn alloy.

The present study conducted a cross-sectional analysis of the Kriab samples, and the findings are presented in Figure 7. The mapping picture revealed that the signal of each element does not follow the border line between the reflective coating and glass layers, as observed for Si and O (Figure 7a,b), which should represent SiO_2_ present in the glass layer. This discrepancy can be attributed to the charging effects that are typically observed in non-conductive and uncoated samples during scanning electron microscopy (SEM) imaging. In our study, the Kriab samples were not coated with gold prior to examination. Kanaya et al. [21] previously reported that the electron beam dose can generate a surface electric field, leading to SEM image perturbations.

Moreover, an uneven distribution of tin clusters was detected within the lead metal layer, as evidenced by the cross-sectional X-ray mapping depicted in Figure 7a,b, which is in agreement with the X-ray diffraction (XRD) results obtained from Kriab2 and Kriab3, respectively. The observed inhomogeneity could be related to the distinct preferred orientations observed in the corresponding XRD patterns. Notably, the XRD pattern of the Kriab3 sample indicates that Pb_2_CO_3_Pb(OH)_2_ is the prevailing phase, contrasting with that of the Kriab2 sample, in which lead is the main constituent with only a small quantity of tin. The X-ray mapping outcomes of both samples reveal that the elemental composition of the coating layers is quite comparable, with the exception of potassium (K), which is only present in the Kriab2 sample. In prior research conducted by a group of scholars [22,23], the joint corrosion products of metal/glass from various historical objects were examined, and it was discovered that corroding glass creates alkaline surface films that may lead to unique metal corrosion products in the contact zone. The pH value also has a crucial role in the formation of corrosion products, with the concentration of Na^+^ and K^+^ ions in the glass serving as a key factor in controlling the pH of the corroding layer. It may be postulated that the evenly dispersed K element in the reflective layer of Kriab3 may be responsible for inhibiting the formation of this hydrocerussite phase. An investigation in more detail should be performed to clearly understand this corrosion process, especially in the lead-coated glass.

The Kriab4 sample is the only sample in this study having a significant amount of NaPb_2_(CO_3_)_2_(OH) in the reflective coating layer. Figure 7c presents the corresponding correctional X-ray mapping of this sample, demonstrating the presence of various elements. Notably, the reflective coating of the Kriab4 sample contained a small quantity of Na and Zn clusters surrounded by Pb, C, and O, which can be attributed to the NaPb_2_(CO_3_)_2_(OH) phase detected in the XRD pattern. This phase has been previously identified in the analysis of historical materials, such as hollow blown glass beads that were coated with molten lead alloys internally [23]. The presence of Na and Zn clusters in proximity to the glass surface may be linked to the corroded area. Additionally, the observation of Zn may be associated with the intentional doping of metal alloys into the lead by ancient craftsmen to adjust the melt condition before mirroring. Furthermore, calcium (Ca) and magnesium (Mg) were found near the corroded area and could be attributed to Rak Samuk, which is a composite of lacquer and burnt brick powder, as mentioned in the first paragraph of this section.

Table 3 presents the elemental composition of the reflective coating and the oxide composition of the glass layer in the selected ancient Kriab mirrors as determined by WDS. It is intriguing to note that no Sn was detected on the reflective coating of the Kriab2 sample, despite the presence of quite distinct XRD peaks in its corresponding XRD patterns. The inhomogeneous distribution of the Sn element found in its corresponding X-ray mapping (Figure 6) may be the underlying cause of this anomaly. It is worth noting that, due to the random selection of only three points on the reflective coating area for WDS analysis, the Zn cluster area may not have been included in the chosen area of analysis. Furthermore, the presence of Ca in the WDS and X-ray mapping results in this Kriab2 sample, could potentially be attributed to the use of a refractory crucible for melting lead in ancient times, or the intentional addition of Ca as alloying elements by skilled craftsmen, to enhance the mechanical properties of the lead casting. Significantly, the reflective coatings of Kriab5 and Kriab6 samples showed detectable amounts of boron (B) in addition to lead and other elements. This could be attributed to the use of borax, which has been widely used as a flux for various metal alloys since ancient times. Notably, these two samples are from the same period and antiquity.

The oxide compositions of the glass layer in various colors of the ancient Kriab-mirror samples, as determined through WDS analysis, are presented in Table 3. Boron oxide was detected in the colorless, red, and green samples, confirming the use of borax by ancient Thai glassmakers during the glass melting process. Previous studies [2,6] did not identify boron oxide, but this could be due to limitations in the EDS technique or the absence of B_2_O_3_ in their samples.

Interestingly, the amount of PbO varied in each glass sample. The blue Kriab6 sample contained a high amount of PbO despite the absence of B_2_O_3_, while Kriab5, from the same antiquity, had a relatively high amount of B_2_O_3_. This discrepancy in the glass composition may reflect differences in ancient glass workshops, or ancient glassmakers may have empirically developed their own unique glass recipes. While anecdotal evidence suggests the existence of two or three glass workshops during the mid-19th century, no strong evidence has been found apart from an antique book mentioned in the introduction. Of note, the glassmaker of the oldest green Kriab7 sample (mid-18th century) in our study, which had the highest amount of SiO_2_ (≈77 wt%), may have used a particularly efficient furnace capable of melting glass at higher temperatures.

Numerous studies have investigated mosaic glass tesserae from the Roman and Byzantine periods, which have been discovered in many historic sites and artifacts throughout Europe and Western Asia [24,25,26,27,28]. Ancient mosaic glass can be either transparent or opaque. Since glasses are typically amorphous and transparent, the presence of heterogeneity in the glass matrix due to phase separation or crystallization can cause it to become opaque, depending on the size of the crystals. In cases of crystallization, some experts prefer to use the term “glass-ceramic”. Many of the opaque mosaic glass tesserae that have been examined are, in fact, glass ceramics [25].

The initial translucent mosaic glass tesserae that existed in ancient times were manufactured using natron or soda-lime silicate. This particular type of glass is distinguished by its low levels of potassium, magnesium, and phosphorus. In the Islamic world, plant ash rich in soda gradually replaced natron, which had originated in Egypt, while central and northern Europe started using wood ash rich in potassium. During the same period, highly leaded glasses emerged in Europe and the Islamic East [9,10,29].

Our Blue Kriab6 glass has a composition that closely resembles one of the blueish glass tesserae cited in reference [29]. The mirror sample in the reference was identified as a soda-ash lead glass with a high PbO content ranging from 40 wt% to 50 wt%, a Na_2_O content of about 5 wt% to 7 wt%, and very low CaO levels ranging from 0.04 wt% to 1.0 wt%. This type of glass is commonly referred to as Islamic lead glass and was first characterized as soda silica glass with lead contents between 33 wt% and 40 wt% by Sayre and Smith [30]. In the Kriab6 sample, the higher amount of PbO (59 wt%) than in previous works may have been intentionally added by the Thai glassmaker to reduce the melting temperature of the glass. The presence of Fe_2_O_3_ impurities may be attributed to the silica source, which could have been sand or rice husks. The 2 wt% copper in the glass may be residual colorants or part of the lead resource.

Although evidence of high-boron Byzantine glasses has been found in many mosaic tesserae, only small amounts ranging from 400 to 2070 ppm (μg/g) have been reported in reference [28]. The high amount of borax in our Kriab2, Kriab5, and Kriab7 samples, approximately 12 wt% to 16 wt%, remains unclear and makes it difficult to classify the type of glass according to previous works. It is possible that the Thai glassmaker intentionally added a high amount of boron to reduce the viscosity of the glass batch, making it easier to form glass sheets by blowing as a large globe or drawing a thin sheet of glass, which is still unknown. The thin glass layer of the ancient Kriab-mirror tesserae, not more than 1 mm thick, may explain the relatively weaker color compared to thicker glass layers typically found in Roman and Byzantine mosaic tesserae, some of which are opacified. To intensify the hues of their glass, ancient Thai glassmakers may have used lead coating to reflect and enhance the colors. This unique and rare invention makes Kriab-mirror tesserae particularly noteworthy.

The presence of K_2_O in the ancient Kriab-mirror samples can be attributed to the use of saltpeter or potassium nitrate, which were commonly used in glass recipes, as described in an antique book written by a Thai Royal glassmaker [1]. Saltpeter was valued for its purity and effectiveness as an oxidizing agent, making it ideal for melting lead glass without reducing the lead oxide to metallic lead. It was also used in British lead-crystal glass, and historical evidence indicates that the East India Company imported it from India, potentially facilitating knowledge transfer between British, Indian, and Thai glassmakers [31]. Further research in collaboration with the Myanmar government to study glass mosaics in Myanmar could provide additional insights into the use of lead-crystal glass in ancient production, as Myanmar and India were part of the British Empire.

### 3.2. Adhesive Property of the Alternative Kriab Mirrors

The modified tape casting method was employed to fabricate alternative Kriab mirrors with a lead alloy coating. The temperature control during the coating process had a significant impact on the hydroxyl groups present on the silica surface. These groups act as major adsorption sites, which can either be “freely vibrating” or enhanced by pretreatment temperature [32]. Although it is difficult to directly measure the number of adsorption sites on the glass surface while hot, we indirectly evaluated this by performing single-lap-joins under static tensile tests on selected alternative Kriab samples.

Figure 8 displays the relationship between the shear strength of the adhesive bonding steel plates to epoxy resin and the corresponding shear strain values. The mean shear strength is determined to be 11.215 MPa. The observed linear relationship between shear strength and shear strain is consistent with prior findings. The failure of the single epoxy resin specimen is depicted in Figure 9. The adhesive residue was detected on both sides of the steel plates, indicating a cohesive failure (CF) mode, or an adhesive failure, as has been reported in previous studies [33,34]. This could be attributed to the non-uniform mixing of the adhesive during specimen preparation. Similar results were obtained from the shear stress-strain curves of alternative Kriab mirrors, as demonstrated in Figure 10. The shear strength of the alternative Kriab mirrors relative to the single epoxy resin specimen is depicted in Figure 11. Notably, the blue Kriab mirror exhibited the highest shear strength, approximately 8.867 MPa, which is close to that of the epoxy resin specimen. The failure mode of the blue Kriab mirror (Figure 12) differed from other samples, displaying some peel-off between the glass and metal coating interface. In contrast, other colored Kriab mirrors exhibited failures at the interface between the epoxy layer and glass surfaces. The lower shear strength values for the red, green, and colorless Kriab mirrors are likely due to these types of failures.

Assessing the strength of the adhesive bond between a glass-and-metal interface using a conventional single-lap-joint specimen configuration is a challenging task. Various factors, including overlap length, adherend yield strength, adhesive plasticity, and bond line thickness, can significantly impact joint strength prediction, potentially leading to inaccurate analyses of adhesive behavior. Stress concentrations resulting from these factors can contribute to fracture initiation in adhesive joints, further complicating analysis. To address this issue, researchers have attempted to modify the sample configuration or use numerical determination approaches [35,36,37]. However, a significant challenge in this study is to prepare a suitable specimen that accurately estimates the true adhesive strength between the glass and metal layer, minimizing the potential for errors introduced during specimen preparation and testing.

In our previous study [38], we explored modifications to the specimen configuration from the standard ASTM protocol, as depicted schematically in Figure 13. Our findings revealed that the average maximum shear force between adhesive-bonded steel plates and epoxy resin was 7.73 kN for the ASTM standard single-lap-joint specimens, characterized by the typical 180° shear direction shown in Figure 13A. In contrast, the modified configuration (peel-off 90° shear direction, Figure 13B) exhibited a lower value of 2.95 kN. These results underscore the significance of the direction of applied force in this measurement. To evaluate the modified configuration, we employed the static tensile test method using a modified single-lap-joint specimen setup depicted in Figure 14a. The failure images of the alternative Kriab samples utilizing this modified method, illustrated in Figure 14b, indicate that the lead alloy layer was completely peeled off from the glass surfaces, in contrast to the failure observed in the standard method (Figure 12). Notably, the maximum shear force of one of the alternative Kriab samples utilizing this modified method was as low as 0.98 kN, suggesting a peel-off nature of the glass-to-metal bonds of the alternative Kriab mirrors. To gain a deeper understanding of the behavior of this glass-to-metal interface in the Kriab mirrors and validate the findings obtained through the modified method, further detailed analytical modeling is required. Additionally, to better comprehend the bonding between the glass-to-lead layer, which links to the coating parameters such as the temperature of the glass sheet or other pre-surface treatment, surface analysis approaches such as X-ray photoelectron Spectroscopy (XPS) should be employed to assess the number of adsorption sites, such as hydroxyl groups or other surface defects, on the glass surface.

### 3.3. CIE-Lab Color of the Ancient and Alternative Kriab-Mirror Tesserae

The color compatibility of the ancient and alternative Kriab-mirrors was investigated through color measurements in the CIE-lab system, as illustrated in Table 4. The alternative Kriab mirrors were compared to the ancient Kriab mirrors using L*, a*, and b* values, which respectively represent lightness, the red/green coordinate, and the yellow/blue coordinate. The resulting ∆L*, ∆a*, and ∆b* values in Figure 15 and Table 4 can be either positive or negative, with a positive value indicating that the sample is lighter, redder, and yellower and a negative value indicating the darker, greener and bluer than the ancient sample, respectively. The analysis reveals that the color of the alternative Kriab mirrors and the ancient mirrors differs, with the blue and red alternative Kriab mirrors appearing lighter and the colorless and green Kriab mirrors appearing slightly darker than the ancient ones. Additionally, all the alternative Kriab mirrors were found to be greener than the ancient ones, with the highest difference values found in the blue mirror sample. Furthermore, negative values were found in ∆b* for the colorless and red mirrors, while positive values were found in the blue and green Kriab mirrors. This indicates that the colorless and red alternative Kriab mirrors have a stronger blue tone, whereas the blue and green Kriab mirrors are yellower than their ancient counterparts.

The total color difference ($∆E^{*}$) between the ancient and alternative Kriab mirrors was estimated using Equation (2) [39].
(2)$$\Delta E^{*}=\sqrt{{\left(\Delta L^{*}\right)}^{2}+{\left(\Delta a^{*}\right)}^{2}+{\left(\Delta b^{*}\right)}^{2}}$$
where ΔL*, Δa*, and Δb* represent the differences between the L*, a*, and b* values of the alternative Kriab mirrors and the ancient Kriab mirrors, respectively. The calculated ΔE* value is always positive. The blue Kriab mirrors showed the highest ΔE* values (refer to Table 5), indicating significant color incompatibility. Notably, the ancient Kriab mirrors exhibited color heterogeneity due to bubbles dispersed throughout the glass matrices, leading to measurement errors. Furthermore, the base glass and colorant play an important role in controlling the glass color. The color difference results confirm the difference in the base glass of our alternative Kriab mirror. To adjust the color compatibility of future prototypes, careful control of the glass recipe should be carried out.

### 3.4. %Reflectance and Mirror Effect of the Ancient and Alternative Kriab-Mirror Samples

Our investigation of the mirror effect of the alternative Kriab and ancient Kriab-mirror tesserae involved measuring the reflectance of selected colorless, blue, green, and red samples in the visible part region. The reflectance spectra of these samples are presented in Figure 16. We also visually compared the ancient Kriab-mirror tesserae with the alternative Kriab-mirror tesserae in a range of colors, as shown in Figure 17. The front and back sides of the tesserae are depicted in Figure 17a,b, respectively, to highlight the differences between the two types of tesserae in terms of color, texture, and overall appearance.

It is important to note that the ancient and alternative mirror samples we chose to measure the reflectance are different from the ones used in the CIE lab color measurement. This was necessary because the samples used for CIE-Lab color measurement were utilized for SEM and WDS analysis, requiring the use of different samples for reflectance measurement. Our aim was to demonstrate the variety of color shades and tones in the mirror samples, and this selection allowed us to achieve that objective while also enabling a more comprehensive investigation of the tesserae.

In Figure 16, the reflectance spectra captured from the green Kriab mirror, both ancient and alternative ones, clearly exhibit prominent peaks at around 500 nm, which are indicative of Cu^2+^ blue–green glass or glaze. This finding is consistent with earlier studies on the optical characteristics of copper-containing materials [40,41], which suggest that the electronic transitions associated with Cu^2+^ ions can give rise to reflection in the blue–green region of the visible spectrum. For the blue Kriab mirror samples, both ancient and alternative samples display three well-defined cobalt (Co^2+^) bands at approximately 530–540, 590–600, and 650–670 nm [39,40,41,42], but the alternative Kriab sample has higher %reflectance than that of ancient one. The red Kriab mirrors of both ancient and alternative ones exhibit distinct absorbance bands that correspond to the surface plasmonic resonance (SPR) of the copper nanoparticles, at around 565 nm, and to isolated Cu^0^ atoms, at approximately 430 nm [41,43,44,45].

The overall %reflectance in the visible part (380–700 nm) of ancient and alternative Kriab mirrors indicates that all mirror samples exhibit lower %reflectance of less than 60% for the colorless old and new mirrors compared to standard silvering mirrors (as seen in the reference in Figure 16). This low %reflectance is consistent with the appearance of all old and new Kriab samples shown in Figure 17. Each alternative Kriab mirror shows more or less discrepancy when compared to the ancient ones. Despite this, professional conservators accept the color incompatibility of the alternative Kriab mirrors and have used them in a restoration project, as shown in Figure 18. Objects with previously used different colored Kriab for decoration were chosen to ensure that the color incompatibility did not affect the overall look of the restoration work. These alternative Kriab mirrors performed better in this restoration project compared to commercial silvering mirrors, which have excessive brightness. However, more compatible alternative or replicated Kriab mirrors are still necessary for antiquities that require compatibility in restoration work.

Another benefit of the alternative Kriab-mirror tesserae is that they are thinner than commercially available mirrors, as shown in Figure 17. The thicker glass of commercial mirrors makes restoration work more challenging, as the surface of the antiquity needs to be polished to ensure the same level as the newly added mirror. The unique characteristics of these alternative Kriab mirrors also make them ideal for creating new contemporary fine art objects, as they are thin and easily cut into small pieces.

### 3.5. Weathering Resistance of the Alternative Kriab Mirrors

Glass-based historical artifacts degrade in a variety of ways depending on their composition and the environment to which they were exposed. Rainwater and environmental humidity regulate the alteration mechanisms of glasses exposed to outdoor and indoor environments [46]. The degradation of the alternative Kriab specimens was assessed in this work utilizing a QUV accelerated experiment to determine their weathering durability. It has been observed that the weathering resistance of these mirrors is low, as evidenced by the visible effects on translucent glass surfaces after a mere 14 days, as shown in Figure 19. Furthermore, the %reflectance across the visible spectrum (approximately 400 to 800 nm) and extending up to ~1000 nm has been examined, as depicted in Figure 20. Following 14 days of weathering tests, the findings indicate a general reduction in %reflectance across all glass surfaces, with the most significant changes occurring in the colorless and blue glasses. However, the green and red color glasses exhibited only minor differences in %reflectance.

Figure 21 and Table 6 provide evidence of the degradation of glass surfaces resulting from the leaching of alkaline and lead elements due to hydrolytic attack on the less chemically stable phase. The PbO-B_2_O_3_-SiO_2_ system exhibits a two-liquid phase separation [47], and all glass compositions in Table 1 fall within the ternary phase diagram’s two-liquid area. After a 14-day QUV test, the green glass surface revealed evidence of phase separation between the affected (translucent surface, spectrum 1) and unaffected area (clear surface, spectrum 2). The affected area had higher alkaline sodium and lead contents than the unaffected area. Glass compositions with high lead content are well known for their susceptibility to moisture absorption [46]. Incorporating lead oxide into glass batches lowers their melting temperature; however, concentrations exceeding 40 wt% may introduce bonds [-O-Pb-O] into the glass structure, decreasing its chemical stability [48,49]. The attack or pitting typically appears isolated at various surface points but may generate an interconnected, continuous alteration layer as it grows. Lixiviated alkaline from the glass can interact further with environmental gases such as CO_2_, SO_2_, and NO_x_, inducing salt crystallization on the glass surface and fissures inside the pits, leading to detachment [50].

## 4. Conclusions

In conclusion, this study provides valuable insights into the characteristics of both ancient and alternative Kriab mirrors, shedding light on the unique recipe of glass mosaic mirror tesserae found in Thailand from the 18th to 19th century. Our analysis of XRD diffraction patterns and X-ray mapping of reflective coatings has revealed corroded compounds caused by alkaline surface films at the glass-to-metal interface. The thinness and low reflectance of the alternative Kriab mirrors make them more suitable for restoration work and even the creation of contemporary fine art objects.

However, we have also identified poor weathering resistance in the lead-based glass used in the alternative Kriab mirrors, indicating the need for further research to develop more stable glass formulations and lead-free alloys as alternative coatings to address environmental concerns. These findings are of significance not only for the conservation of Kriab mirrors but also for historical objects of lead-coated glasses found worldwide. They can guide future experiments for the production of new prototypes with better color compatibility and also assist professional conservators in their efforts to restore and preserve valuable artifacts.

## Figures and Tables

**Figure 1 materials-16-03321-f001:**
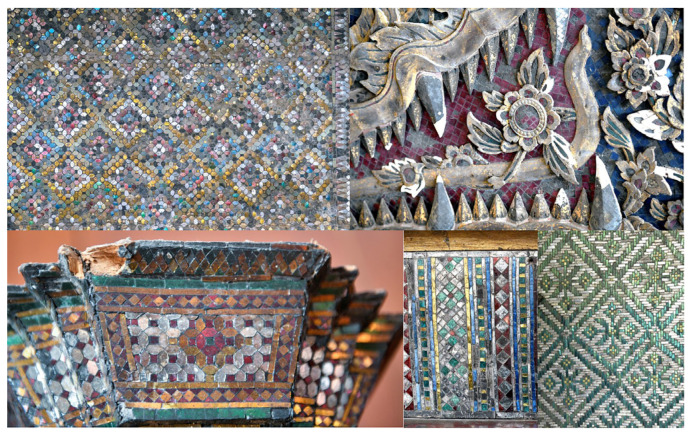
The mosaic arts of the ancient Kriab-mirror tesserae from temples and the National Museum in Thailand’s central region.

**Figure 2 materials-16-03321-f002:**
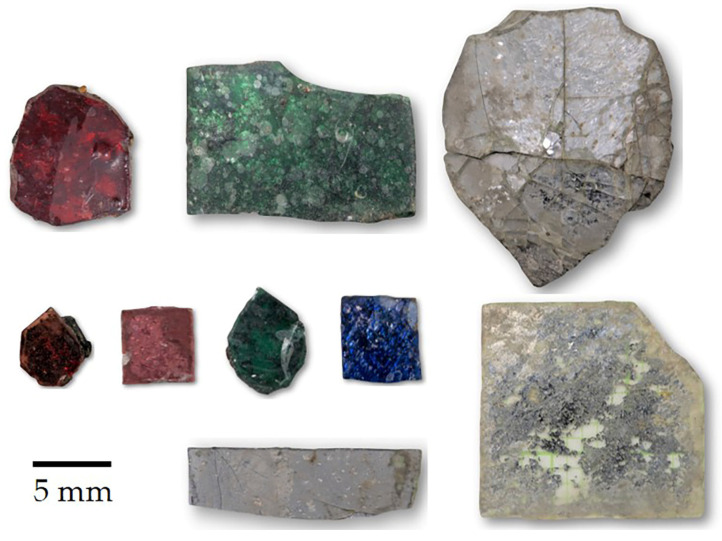
Pieces of the ancient Kriab-mirrors tesserae in various colors.

**Figure 3 materials-16-03321-f003:**
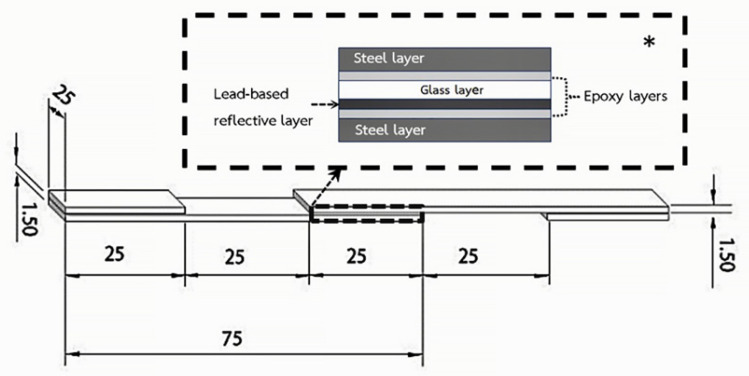
Specimen configuration of single-lap-joint setup specimen (dimensions in mm) following ASTM D1002-99 standard. * The enlargement of the cross-sectional area of the mirror sample and metal sheet is shown in the inset (the thickness of each layer is not present in the realistic ratio).

**Figure 4 materials-16-03321-f004:**
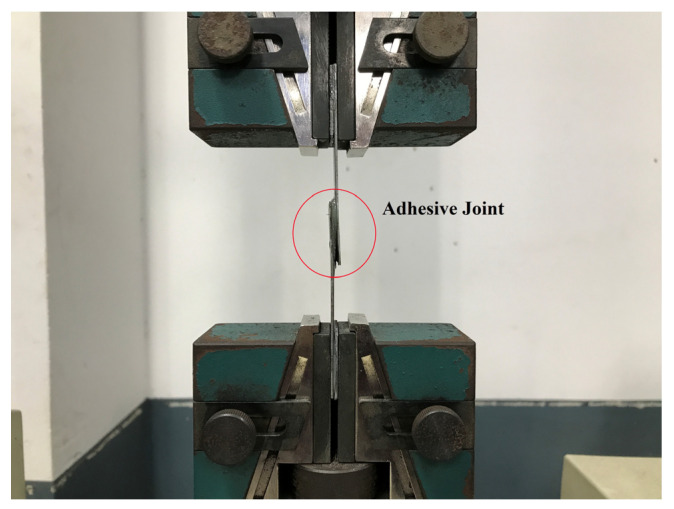
The experimental setup of the ASTM standard single-lap-joint and under static tensile test method of the only epoxy adhesive.

**Figure 5 materials-16-03321-f005:**
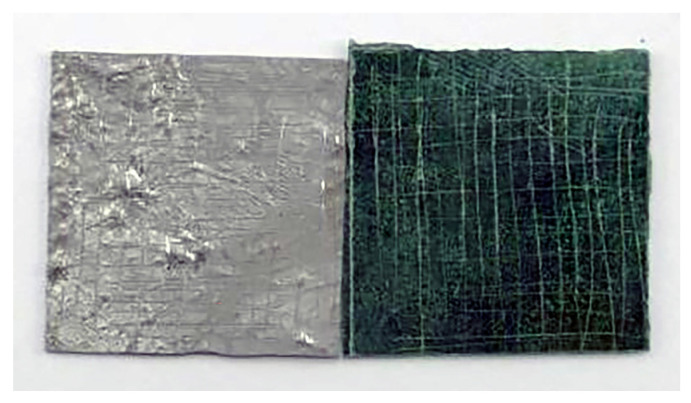
The green Kriab mirrors’ glass and lead-based alloy surfaces scratched by a high-hardness metal tool.

**Figure 6 materials-16-03321-f006:**
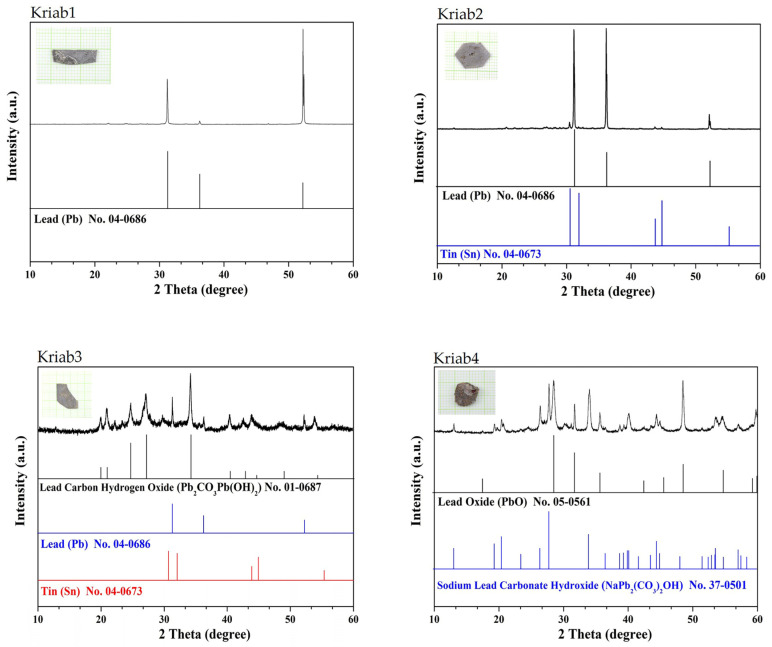
XRD patterns of the reflective coatings on the backs of ancient Kriab mirrors.

**Figure 7 materials-16-03321-f007:**
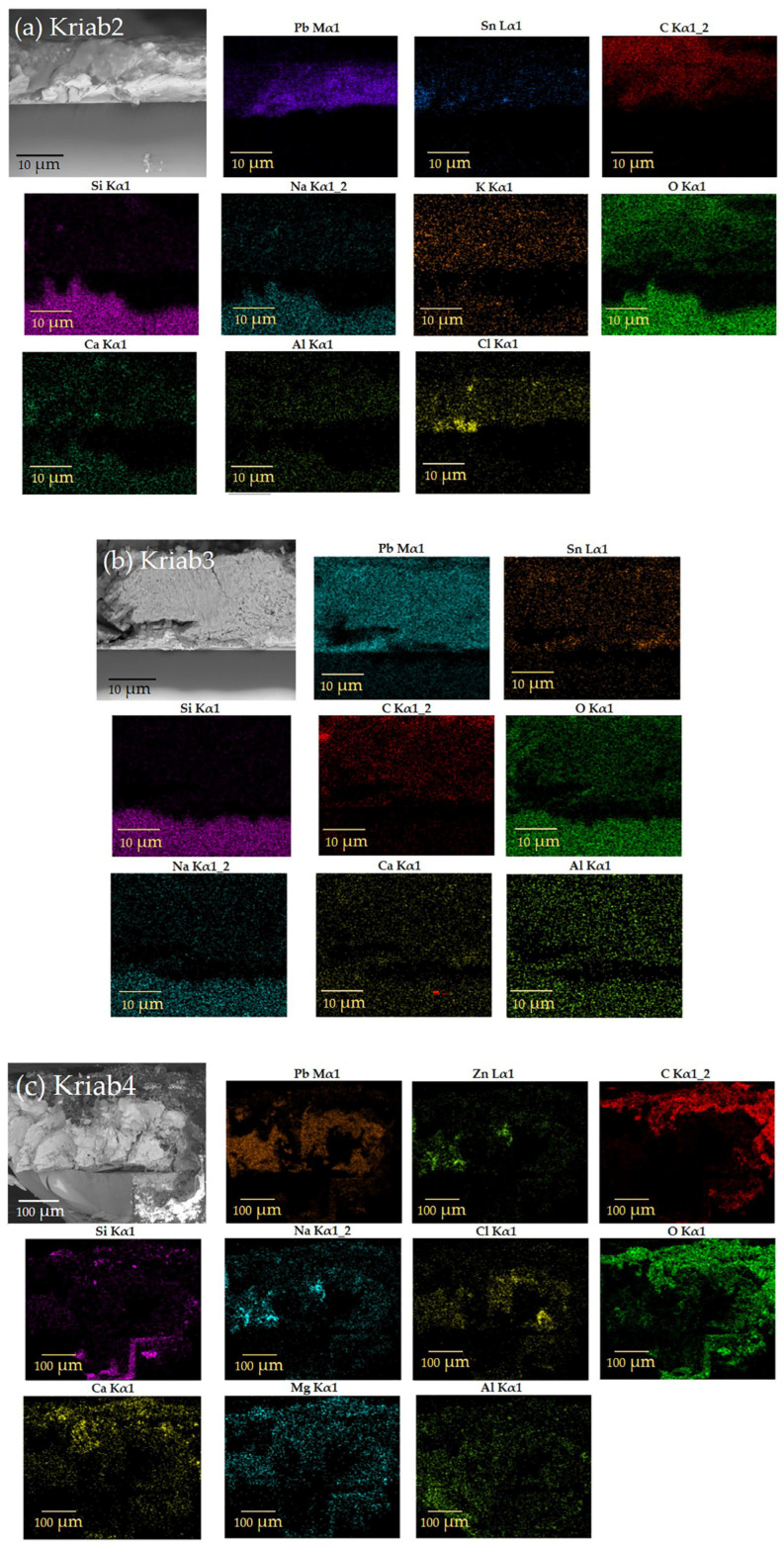
Cross-sectional SEM and XRD mapping of the selected ancient Kriab mirrors: (**a**) Kriab2 and (**b**) Kriab3, and (**c**) Kriab4.

**Figure 8 materials-16-03321-f008:**
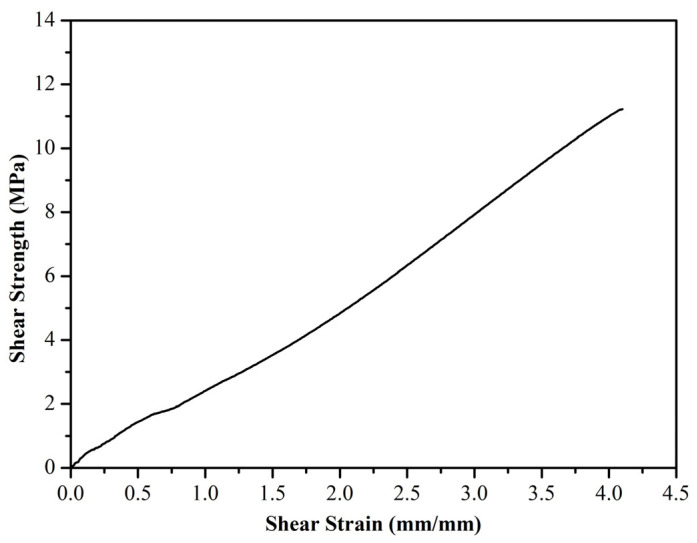
Shear stress–strain curve of the only epoxy adhesive specimen.

**Figure 9 materials-16-03321-f009:**
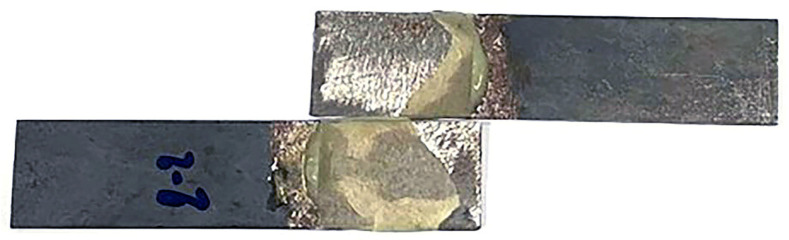
Failure of the only epoxy adhesive specimen.

**Figure 10 materials-16-03321-f010:**
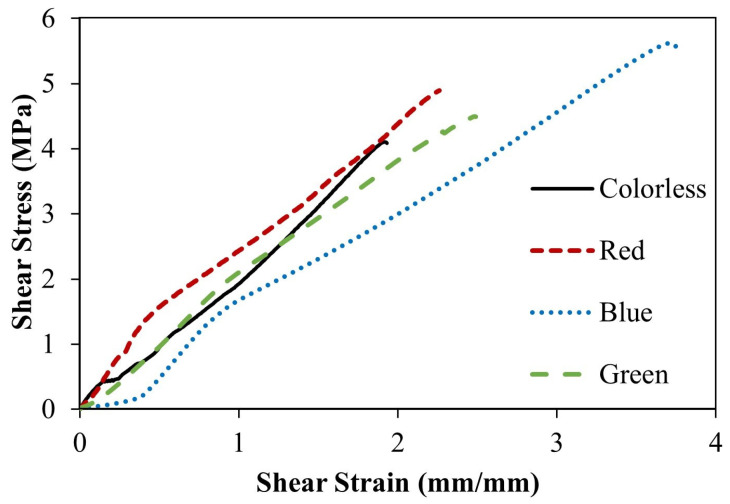
Shear stress–strain curves for the alternative Kriab mirrors in various colors.

**Figure 11 materials-16-03321-f011:**
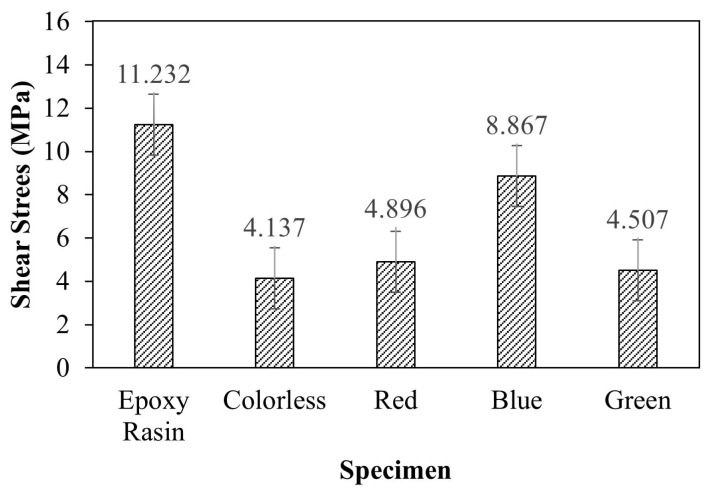
Comparison of the shear strength value of various colored alternative Kriab mirrors with that of the only epoxy resin specimen.

**Figure 12 materials-16-03321-f012:**
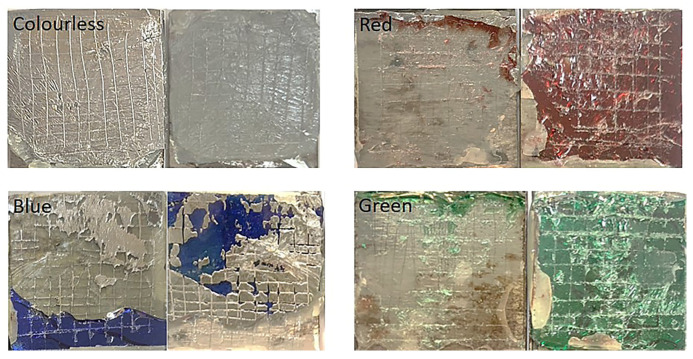
Failure images of the alternative Kriab mirror from the ASTM standard single-lap-joint method.

**Figure 13 materials-16-03321-f013:**
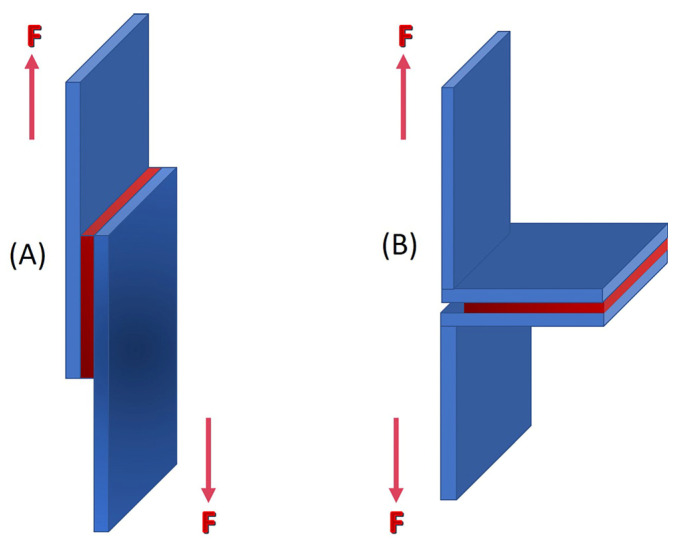
Schematic presentation of (**A**) the standard single-lap-joint and (**B**) the modified single-lap-joint specimen configuration.

**Figure 14 materials-16-03321-f014:**
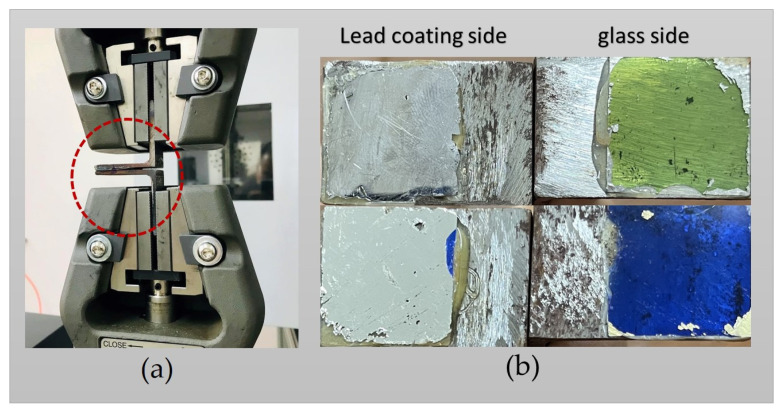
(**a**) The modified single-lap-joint specimen set up under the static tensile test method and (**b**) Failure images of the alternative Kriab mirrors from this modified method.

**Figure 15 materials-16-03321-f015:**
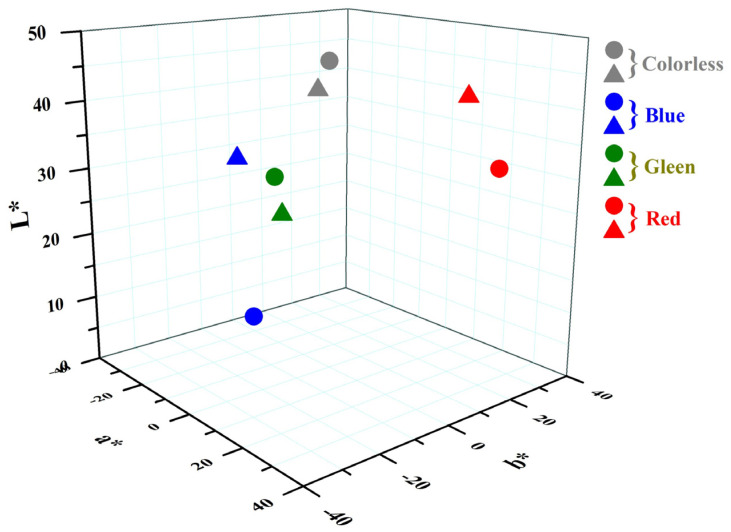
Comparison of ancient and alternative Kriab mirrors: Ancient Kriab mirrors are represented by circles, while alternative mirrors are denoted by triangles.

**Figure 16 materials-16-03321-f016:**
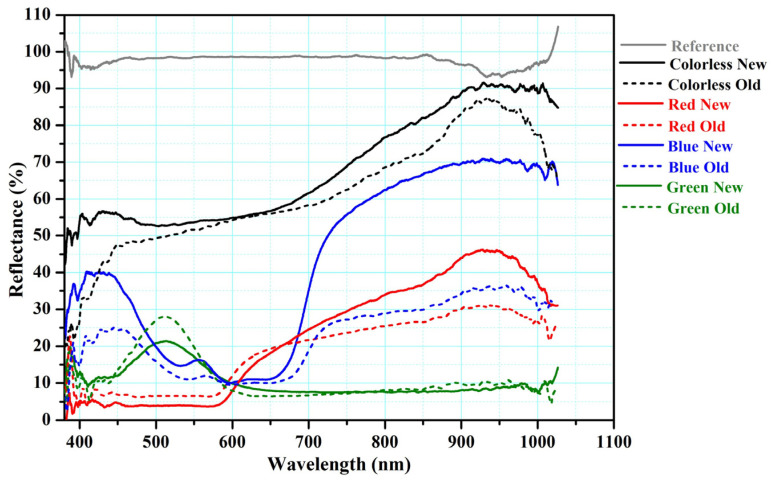
Reflectance of colored Kriab mirrors compared with the standard silvering mirrors as reference.

**Figure 17 materials-16-03321-f017:**
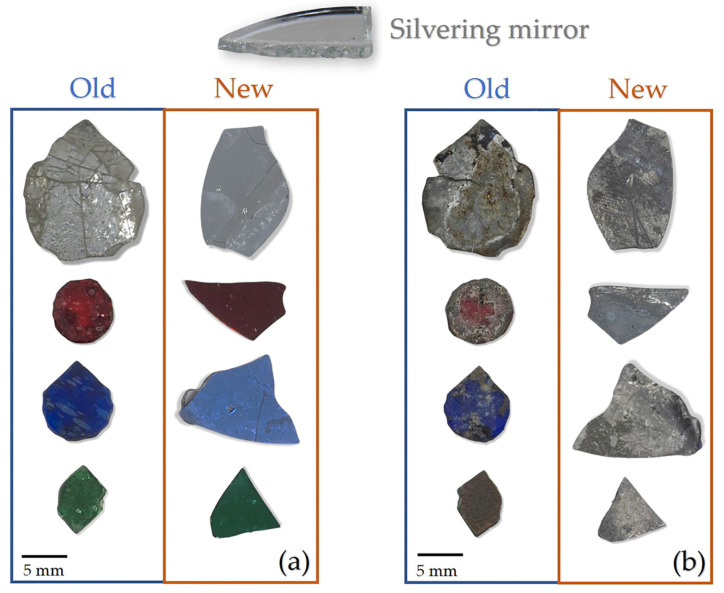
Comparison between ancient (old) Kriab-mirror tesserae and alternative (new) Kriab-mirror tesserae with a variety of colors. (**a**) Front side of the tesserae, and (**b**) Back side of the tesserae. A piece of commercial silvering mirror is also compared.

**Figure 18 materials-16-03321-f018:**
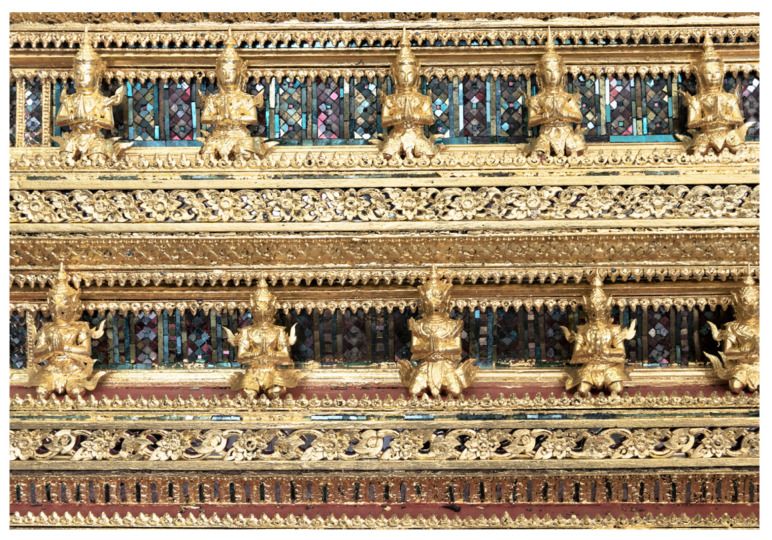
Restoration work conducted on antiquity by Thailand’s Ministry of Culture’s Office for Traditional Arts and Department of Fine Arts specialists.

**Figure 19 materials-16-03321-f019:**
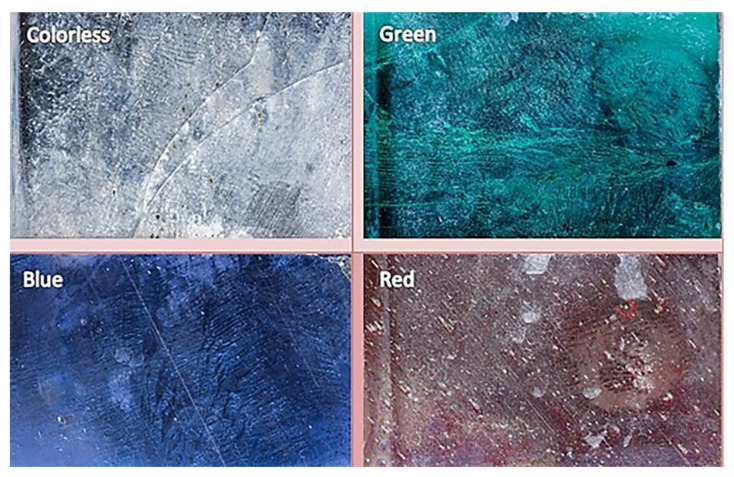
Surfaces of the alternative Kriab mirror of various colors after being subjected to a QUV accelerated weathering test for 14 days.

**Figure 20 materials-16-03321-f020:**
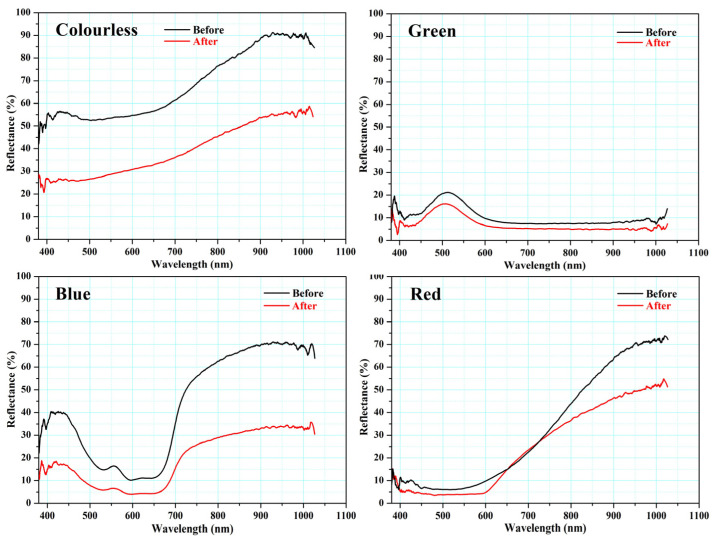
Comparison of the %reflectance of various colors of alternative Kriab mirrors before and after a 14-day QUV accelerated weathering test.

**Figure 21 materials-16-03321-f021:**
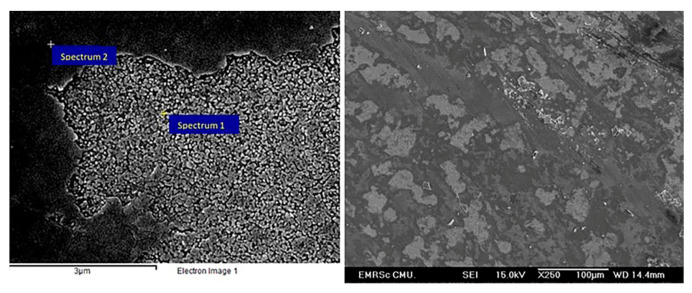
Micrographs of the alternative green Kriab mirror’s surface after being subjected to a QUV accelerated weathering test for 14 days.

**Table 1 materials-16-03321-t001:** The composition of the investigated color glass samples in wt% by SEM-EDS analysis.

	Based Glass Composition				Colorants and Possible Impurities			
Oxides	SiO_2_	B_2_O_3_	PbO	Na_2_O	Al_2_O_3_	CuO	CoO	SnO_2_
Colorless	23.23	21.63	44.87	6.65	1.19	2.43	-	-
Blue	29.74	18.68	42.44	5.85	1.29	2.01	-	-
Green	23.64	17.40	48.70	5.87	1.40	3.00	-	-
Red	24.67	20.23	48.70	5.54	0.87	0.02 *	-	0.09 *

* EDS analysis by SEM (JSM-IT800, Japan Electron Optics Laboratory Co., Ltd. (JEOL), Tokyo, Japan).

**Table 2 materials-16-03321-t002:** Scotch-Weld^TM^ Epoxy Adhesive DP100 Plus Clear material properties [19].

Manufacturer Typical Neat Resin	Properties
Color	Clear
Hardness (ASTM D2240) Shore D	65–70
Work-life	3–4 min
Tack-free Time	9–10 min
Time to Handling Strength	20 min at 23 °C
Cure Time	48 h at 23 °C
Elongation	75%
Tensile Strength	12.8 MPa

**Table 3 materials-16-03321-t003:** Elemental and oxide composition of the reflective coating (metal) layer and glass layer, respectively, of the selected ancient Kriab mirrors, determined by Wavelength Dispersive Spectroscopy (WDS).

	Weight%	Kriab2 (Colorless)	Kriab5 (Red)	Kriab6 (Blue)	Kriab7 (Green)
**Element (in metal layer)**	O	10.754	6.517	4.435	N/A
	B	-	0.295	1.850	
	Pb	72.686	86.545	89.589	
	Sn	-	6.405	-	
	Bi	-	-	-	
	Na	-	0.238	-	
	K	7.550	-	-	
	Ca	3.825	-	-	
	S	5.185	-	-	
	C	-	-	4.126	
	**Period**	**mid-19th** **century**	**mid-19th** **century**	**mid-19th** **century**	**mid-18th** **century**
**Oxide (in glass layer)**	SiO_2_	50.420	43.733	31.405	76.853
	B_2_O_3_	14.950	16.108	-	12.377
	PbO	9.115	32.551	59.962	7.902
	Na_2_O	4.956	1.776	4.315	-
	K_2_O	3.795	1.237	-	1.337
	MgO	0.274	0.253	-	0.019
	CaO	11.936	3.269	-	0.038
	Al_2_O_3_	2.676	1.008	0.654	0.262
	Fe_2_O_3_	1.429	0.065	1.404	0.216
	CuO	-	-	2.259	0.847
	ZnO	-	-	-	0.150
	MnO	0.448	-	-	-

**Table 4 materials-16-03321-t004:** Color coordinates of the ancient Kriab mirrors compared to that of the alternative Kriab mirrors.

Ancient Kriab	CIE-Color Coordinates	Alternative Kriab	CIE-Color Coordinates	Difference
** 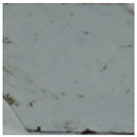 **	L* = 46.342 a* = −0.097 b* = −0.055	* 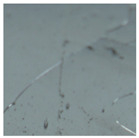 *	L* = 42.007 a* = −3.211 b* = −0.852	∆L* = −4.335 ∆a* = −3.114 ∆b* = −0.797
** * 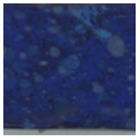 * **	L* = 15.156 a* = 11.891 b* = −30.716	* 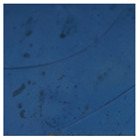 *	L* = 32.646 a* = −11.874 b* = −17.412	∆L* = 17.490 ∆a* = −23.765 ∆b* = 13.304
** * 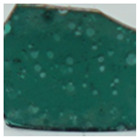 * **	L* = 27.521 a* = −18.599 b* = −1.684	* 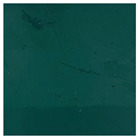 *	L* = 21.185 a* = −19.225 b* = 0.903	∆L* = −6.336 ∆a* = −0.626 ∆b* = 2.587
** 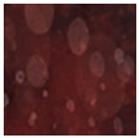 **	L* = 33.569 a* = 36.618 b* = 16.547	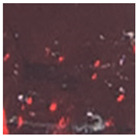	L* = 43.146 a* = 31.205 b* = 12.094	∆L* = 9.577 ∆a* = −5.413 ∆b* = −4.453

**Table 5 materials-16-03321-t005:** Color differences between the ancient Kriab mirrors compared to that of the alternative Kriab mirrors.

Color of Mirrors	∆L*	∆a*	∆b*	∆E*
Colorless	−4.335	−3.114	−0.797	5.4
Blue	17.490	−23.765	13.304	32.4
Green	−6.336	−0.626	2.587	6.9
Red	9.577	−5.413	−4.453	11.9

**Table 6 materials-16-03321-t006:** Elemental compositions of two EDS spectra of the alternative green Kriab mirror’s surface.

Spectrum 1 (Translucent Surface)			Spectrum 2 (Clear Surface)		
Element	Weight%	Atomic%	Element	Weight%	Atomic%
C K	14.35	35.90	C K	23.88	54.57
O K	22.32	41.93	O K	16.11	27.19
Na K	2.63	3.44	Na K	1.83	2.19
Si K	10.72	11.47	Si K	9.35	9.13
Pb M	49.98	7.25	Pb M	48.83	6.47

## Data Availability

Not applicable.

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
