# Peer review of "Chemical and Mechanical Characterization of the Alternative Kriab-Mirror Tesserae for Restoration of 18th to 19th-Century Mosaics (Thailand)"

_materials, 2023, doi:10.3390/ma16093321_

Round 1
Reviewer 1 Report
Comments on Manuscript
Title: “Characterization of the replicated Kriab-mirrors for conservative purposes”
1) Line 61 and Line 67 are replicates.
2) Line 73-74: In light of the availability of high-temperature furnaces today (w.r.t the statement on Line 71), justify the use of higher amounts (>50%) of a toxic chemical- PbO in the replicated mirrors, especially in a handcrafted approach where the same comes in direct contact risking the health.
3) Line 76-78: The statement raises critical concerns about the scalability of the approach.
4) Line 400: Mentioning the study used first-generation prototypes with a furnace having a maximum temperature of 900oC, how the temperature range of 900 to 1200oC stated in Line 80 was achieved?
General Feedback:
· The authors may improve the ‘Introduction’ of the manuscript to make it clearer and more organized. Several vital points related to the scope and details of the work are scattered throughout the manuscript.
· The description of the materials and process involved in the replication of the mirrors should be ideally written in the ‘Materials and Methods’ section of the manuscript. For better organization, Section 2.2 which contains information regarding the composition is recommended to be combined with the above.
Author Response
Response to reviewer’s comments
Manuscript ID: materials-2282886
Title: Characterization of the replicated Kriab-mirrors for conservative purposes
Dear Editor of Materials
We are grateful for the opportunity to submit our manuscript, “Characterization of the replicated Kriab-mirrors for conservative purposes", to your esteemed journal. We appreciate the time and effort invested by the reviewers in providing insightful comments and suggestions, which have greatly improved the quality of our work.
We have carefully addressed all the issues the reviewers raised in our manuscript's revised version. Specifically, we have made the changes in the following table.
Thank you again for considering our work, and we look forward to hearing from you soon.
Best regards,
Assoc. Prof. Dr. Kamonpan Pengpat (corresponding author)
Department of Physics, Faculty of Science,
Chiang Mai University, Chiang Mai, 50200, Thailand
Tel: +6653943376
Fax: +6653357512
Email: kamonpan.p@cmu.ac.th
Fax: +6653357512
kamonpan.p@cmu.ac.th
Responses to reviewer #1’s comments
No. |
Comment from referee |
Responses |
1. |
Line 61 and Line 67 are replicates. |
We have already rewritten the introduction part and corrected all mentioned errors. |
2. |
Line 73-74: In light of the availability of high-temperature furnaces today (w.r.t the statement on Line 71), justify the use of higher amounts (>50%) of a toxic chemical- PbO in the replicated mirrors, especially in a handcrafted approach where the same comes in direct contact risking the health. |
Thanks a lot for this concern; we have already addressed this issue in section 2.2 (lines 160-163) and the conclusion section (lines 531-535). |
3. |
Line 76-78: The statement raises critical concerns about the scalability of the approach. |
We have moved this part into section 2.2, which has changed to “Production of replicated Kriab mirrors. The scalability of the approach has been explained in the first paragraph of this section (lines 157-160). |
4. |
Line 400: Mentioning the study used first-generation prototypes with a furnace having a maximum temperature of 900oC, how the temperature range of 900 to 1200oC stated in Line 80 was achieved? |
We have already explained this point in section 2.2 (lines 149-151). |
5. |
The authors may improve the ‘Introduction’ of the manuscript to make it clearer and more organized. Several vital points related to the scope and details of the work are scattered throughout the manuscript. |
I have already rewritten the introduction and organized the essential points more clearly, as demonstrated in Section 1, Introduction (lines 49-122). |
6. |
The description of the materials and process involved in the replication of the mirrors should be ideally written in the ‘Materials and Methods’ section of the manuscript. For better organization, Section 2.2 which contains information regarding the composition is recommended to be combined with the above. |
We have already rewritten section 2.2, which has changed to “Production of replicated Kriab mirrors. All concerned points have been explained (lines 145-186). |

Reviewer 2 Report
1. What is the main question addressed by the research? The main question is related to the physical and mechanical properties of replicated mirrors. 2. Do you consider the topic original or relevant in the field? Does it address a specific gap in the field? The topic is very interesting and different from regular papers on glasses. The gap is replication of piece of arts. 3. What does it add to the subject area compared with other published material? The topic is very interesting and different from regular papers on glasses, as it deals with replication of piece of arts. 4. What specific improvements should the authors consider regarding the methodology? What further controls should be considered? No comments. 5. Are the conclusions consistent with the evidence and arguments presented and do they address the main question posed? Yes. 6. Please include any additional comments on the tables and figures.No comments.
The authors should provide more recent references.
Author Response
Response to reviewer’s comments
Manuscript ID: materials-2282886
Title: Characterization of the replicated Kriab-mirrors for conservative purposes
Dear Editor of Materials
We are grateful for the opportunity to submit our manuscript, “Characterization of the replicated Kriab-mirrors for conservative purposes", to your esteemed journal. We appreciate the time and effort invested by the reviewers in providing insightful comments and suggestions, which have greatly improved the quality of our work.
We have carefully addressed all the issues the reviewers raised in our manuscript's revised version. Specifically, we have made the changes in the following table.
Thank you again for considering our work, and we look forward to hearing from you soon.
Best regards,
Assoc. Prof. Dr. Kamonpan Pengpat (corresponding author)
Department of Physics, Faculty of Science,
Chiang Mai University, Chiang Mai, 50200, Thailand
Tel: +6653943376
Fax: +6653357512
Email: kamonpan.p@cmu.ac.th
Response to reviewer #2’s comments
No. |
Comment from referee |
Response |
1. |
The authors should provide more recent references. |
Thank you for your suggestion. In the revised version of this manuscript, we have included more up-to-date references for readers to follow. |

Reviewer 3 Report
Ancient Kriab mirrors made in Thailand from thin flat glass coated with molten lead are rarely described and insufficiently researched. However, instead of analysing original glass compositions and lead alloys, modern replicas made from borate (!) glass are studied.
1) CIE measurements found that colours do not match perfectly. This could already be seen with the naked eye from the photographs.
2) The loss of mirrors from the support occurs due to the failure of the organic adhesive with whom they are fixed, not from the failure of the bond between glass and lead. The novel measurement of the shear strength is, therefore, irrelevant to judge the suitability of replicas for restoration.
3) The borate glasses studied do not match historic compositions. They are about to be replaced (l. 404/6.)
On these reasons given and despite the fascinating topic of Kriab mirrors, this study on replicas seems to be only of very limited interest to a wider readership. Therefore, rejection is recommended.
Detailed criticism:
3: here and later (e.g., l. 36, 85, 321, 408). The English language does not use ‘conservative‘ as adjective related to (art) conservation. As new replication mirrors are used, a better term (instead of conservation which is related to original parts) would be ‘restoration‘. Suggestion for title: Characterization of the replicated Kriab-mirrors used for restoration. (correct in line 100).
24: ‚‘significant achievement`. Authors should avoid wording which sounds like self-praise. Simply state that replication was achieved.
24/25. The second sentence of the abstract could be deleted without loss of information
28: delete ‘also‘ (what else do they reflect?)
29. Spelling of ‘color‘ varies throughout manuscript. Choose either Birtish or American English and unify spelling. If you prefer British English, it’s colour and utilise (not utilize, emphasize etc.) and characterisation etc.
29-31: Rephrase: Color parameters, L*, a*, and b* values in CIE-Lab standard were measured, could be derived from the CIE-Lab standard after discovering that they differed for each coloured Kriab mirror (The difference is better explained in the following sentence).
47,48,75,79, 125 etc.: :There must always be a blank space between number and physical unit: 0.7 mm, 0.05 mm, 900 °C, 15 kV, a.s.o. in the whole text.
58: Insert ‘co-author‘ before Chanchiaw
62: If ‘previously stated‘ refers to the sentence before, the whole sentence could be deleted, as no new information is provided. If it refers to something else, a precise reference must be given.
72: The term ‘flux‘ is usually used only for univalent oxides in glass production. Pb(II) is a network stabiliser. Replace ‚‘as a flux‘ by ‘to lower the temperature for fusion‘
77: change?? Replace: vary
78: Does the adhesive strength really vary significantly? For a given colour, this seems not to be true (note the small error bars in Fig. 12).
83: The exact composition of the alloy must be given! Otherwise no one could replicate findings. Has the metal on originals been analysed? It would also be interesting to describe more details of the process. How was the metals spread to a thin layer? Just by letting a small layer solidify in contact to glass and removing excess liqiud alloy by pouring? Does drossing (oxide formation in contact with air) occur (depends on alloy chosen)? Are disturbances of the reflection by oxide particles to be seen under the microscope in originals and replications? (Fig. 2 shows inhomogenities).
92/93: Why is the bonding between glass and metal coating of particular relevance? Loss of original mirrors seems not to occur at the interface of glass and metal (leaving lead on the support) but between the lead and the support. So it is a failure of the organic adhesive used to fix them!
101: Why is colour measurement crucial? At the end you judge by the naked eye that the colour match is not perfect, but sufficient for use.
126: Clearly state that the replication glasses were analysed, not originals! Why were the compositions not estimated from the batch, especially colourants like Co?
141: EDS analysis from with SEM
149, 151: mm²
179: Table 2: psi is not a SI unit and must be converted to MP. 1850 psi = 12.8 MP. By the way, this is in the range of your measured shear strength.
193: 0.63 W/m2/nm: Correct unit? Irradiance?
210: If you have not measured the compositions of the ancient alloys with EDS (why?), you could estimate the composition from XRD if lead and tin are the only crystalline phases present. This would provide at least some valuable information on the originals.
212 The term ‘lead carbon hydrogen oxide‘ is misleading. Better would be lead carbonate hydroxide (‘hydrocerussite‘)
217: Unevenly distributed tin clusters: This is to be expected for Pb-Sn, see ASM Handbook Vol. 9
226 The measured XRD (Fig. 6, bottom right) is not fully explained by the presence of lead and hydrocersussite alone. Did you try to match the other peaks?. Common corrosion products of lead would be cerussite and hydrocerussite. In contact to soda glass, NaPb2(CO3)2(OH) was also observed (ICDD-PDF 37-501, see WHEN GLASS AND METAL CORRODE TOGETHER, IV: SODIUM LEAD CARBONATE HYDROXIDE, available at Researchgate). Are any of these phases present? Or what else?
230: Figure 7a and b: The signal for SiO2 does not follow the border between lead and glass (as seen in the Pb picture). Why??
251: Chanchiaw ??
257-259: To explain a difference of glasses in a physical property with the presence of 0.02 % CuO does not make much sense, when the glasses have much higher differences in other compounds (see Table 1)
287: replicated
300: reproduced
334/5: …and very likelyly no borax at all!
365: What do you mean with ‘isolated CuO atoms‘?
370: Wrong legend for Fig. 15! (This is legend for Fig. 1)
373: Wrong legend for Fig. 16! (This is legend for Fig. 1)
391: Wrong legend for Fig. 17! (This is legend for Fig. 1)
Author Response
Response to reviewer’s comments
Manuscript ID: materials-2282886
Title: Characterization of the replicated Kriab-mirrors for conservative purposes
Dear Editor of Materials
We are grateful for the opportunity to submit our manuscript, “Characterization of the replicated Kriab-mirrors for conservative purposes", to your esteemed journal. We appreciate the time and effort invested by the reviewers in providing insightful comments and suggestions, which have greatly improved the quality of our work.
We have carefully addressed all the issues the reviewers raised in our manuscript's revised version. Specifically, we have made the changes in the following table.
Thank you again for considering our work, and we look forward to hearing from you soon.
Best regards,
Assoc. Prof. Dr. Kamonpan Pengpat (corresponding author)
Department of Physics, Faculty of Science,
Chiang Mai University, Chiang Mai, 50200, Thailand
Tel: +6653943376
Fax: +6653357512
Email: kamonpan.p@cmu.ac.th
Respond to reviewer #3’s comments
No. |
Comment from referee |
Response |
1. |
CIE measurements found that colours do not match perfectly. This could already be seen with the naked eye from the photographs. |
Thank you for your feedback. We appreciate that our prototypes have some colour discrepancies and will aim to improve them in future iterations. In this revised manuscript, we have addressed this issue in the abstract (lines 30-39), section 3.3 (lines 435-439), and conclusions (lines 519-524). |
2. |
The loss of mirrors from the support occurs due to the failure of the organic adhesive with whom they are fixed, not from the failure of the bond between glass and lead. The novel measurement of the shear strength is, therefore, irrelevant to judge the suitability of replicas for restoration. |
Thank you for bringing these issues to our attention. We appreciate your feedback and fully agree with your comments. As such, we have rechecked our data and conducted additional measurements. To clarify this point, we have rewritten the results and discussion section of 3.2 (lines 345-406). |
3. |
The borate glasses studied do not match historic compositions. They are about to be replaced (l. 404/6.) |
We understand that our first prototypes may not yet accurately reflect the historical composition of Kriab mirrors due to limited resources and knowledge. Additionally, previous research on the composition of ancient Kriab mirrors reported the absence of boron due to the limitations of their analytical methods and the fact that their observations were limited to a single location (the Temple of the Emerald Buddha). However, our study investigates Kriab mirrors from various locations and reveals that some of the samples contain a significant amount of boron oxide, providing evidence of the use of borax during ancient times. These new findings could be valuable and could stimulate new discussions among researchers in related fields. |
4. |
On these reasons given and despite the fascinating topic of Kriab mirrors, this study on replicas seems to be only of very limited interest to a wider readership. Therefore, rejection is recommended. |
To address this issue, we have highlighted how our work can attract researchers in related fields through the following points:
|
5. |
3: here and later (e.g., l. 36, 85, 321, 408). The English language does not use ‘conservative‘ as adjective related to (art) conservation. As new replication mirrors are used, a better term (instead of conservation which is related to original parts) would be ‘restoration‘. Suggestion for title: Characterization of the replicated Kriab-mirrors used for restoration. (correct in line 100). |
Thanks for your comment. We agree and have changed the title to “Characterization of the replicated Kriab-mirrors used for restoration” and corrected this point in our revised version. |
6. |
24: ‚‘significant achievement`. Authors should avoid wording which sounds like self-praise. Simply state that replication was achieved. |
We have rewritten the first sentence of the abstract to address this issue, and it has been corrected in our revised manuscript (lines 23-24). |
7. |
24/25. The second sentence of the abstract could be deleted without loss of information |
We have rewritten the abstract to address this issue, which has been corrected in our revised manuscript (lines 23-27). |
8. |
28: delete ‘also‘ (what else do they reflect?) |
We have rewritten this sentence in lines 29-30 (also was deleted). |
9. |
29. Spelling of ‘color‘ varies throughout manuscript. Choose either Birtish or American English and unify spelling. If you prefer British English, it’s colour and utilise (not utilize, emphasize etc.) and characterisation etc. |
We have carefully checked and used British-style spelling in my revised manuscript. |
10. |
29-31: Rephrase: Color parameters, L*, a*, and b* values in CIE-Lab standard were measured, could be derived from the CIE-Lab standard after discovering that they differed for each coloured Kriab mirror (The difference is better explained in the following sentence). |
We have already rewritten this related part in the abstract (lines 30-36). |
11. |
47,48,75,79, 125 etc.: :There must always be a blank space between number and physical unit: 0.7 mm, 0.05 mm, 900 °C, 15 kV, a.s.o. in the whole text. |
We have tried to correct all these errors in our revised manuscript. |
12. |
58: Insert ‘co-author‘ before Chanchiaw |
We have added ‘co-author’ before Chainchiaw in line 79. |
13. |
62: If ‘previously stated‘ refers to the sentence before, the whole sentence could be deleted, as no new information is provided. If it refers to something else, a precise reference must be given. |
We have deleted the ‘previously stated..’ sentence in line 83. |
14. |
72: The term ‘flux‘ is usually used only for univalent oxides in glass production. Pb(II) is a network stabiliser. Replace ‚‘as a flux‘ by ‘to lower the temperature for fusion‘ |
We have corrected this in our revised manuscript. |
15. |
77: change?? Replace: vary |
This was corrected. Note: we have moved this paragraph from the introduction part and rewritten in section 2.2 which has been changed to “Production of replicated Kriab mirrors”. |
16. |
78: Does the adhesive strength really vary significantly? For a given colour, this seems not to be true (note the small error bars in Fig. 12). |
We have rewritten and corrected the results of “the adhesive property of the replicated Kriab mirrors” in section 3.2, and Figure 12 was changed to Figure 11 with corrected error bars. |
17. |
83: The exact composition of the alloy must be given! Otherwise no one could replicate findings. Has the metal on originals been analysed? It would also be interesting to describe more details of the process. How was the metals spread to a thin layer? Just by letting a small layer solidify in contact to glass and removing excess liqiud alloy by pouring? Does drossing (oxide formation in contact with air) occur (depends on alloy chosen)? Are disturbances of the reflection by oxide particles to be seen under the microscope in originals and replications? (Fig. 2 shows inhomogenities). |
We gave the lead alloy composition determined by EDS in section 2.2, lines 168-169. The detail of the production process has been provided in this section. |
18. |
92/93: Why is the bonding between glass and metal coating of particular relevance? Loss of original mirrors seems not to occur at the interface of glass and metal (leaving lead on the support) but between the lead and the support. So it is a failure of the organic adhesive used to fix them! |
To address this issue, we incorporated the modified single-lap-joint from our previous research to demonstrate the peel-off behavior of the glass-to-metal interface. Additionally, this finding can serve as a valuable reference for designing adhesive bonding experiments in future prototypes. Furthermore, we have revised the results about 'the adhesive property of the replicated Kriab mirrors' in section 3.2. |
19. |
101: Why is colour measurement crucial? At the end you judge by the naked eye that the colour match is not perfect, but sufficient for use. |
We have already addressed this issue in section 3.3 (lines 431-438). |
20.
|
126: Clearly state that the replication glasses were analysed, not originals! Why were the compositions not estimated from the batch, especially colourants like Co? |
We have already addressed this issue in section 2.2 (lines 147-157). |
21. |
141: EDS analysis from with SEM |
We have corrected this point in line 185. |
22. |
149, 151: mm² |
We have corrected this point in line 193. |
23. |
179: Table 2: psi is not a SI unit and must be converted to MP. 1850 psi = 12.8 MP. By the way, this is in the range of your measured shear strength. |
We have changed to SI unit as suggested in Table 2. |
24. |
193: 0.63 W/m2/nm: Correct unit? Irradiance? |
We have corrected this point in line 235. |
25. |
210: If you have not measured the compositions of the ancient alloys with EDS (why?), you could estimate the composition from XRD if lead and tin are the only crystalline phases present. This would provide at least some valuable information on the originals. |
We have already rewritten this section and provided the compositions of the ancient alloys from selected Kriab samples in Table 3. |
26. |
212 The term ‘lead carbon hydrogen oxide‘ is misleading. Better would be lead carbonate hydroxide (‘hydrocerussite‘) |
This part has been rewritten in lines 263-270. |
27. |
217: Unevenly distributed tin clusters: This is to be expected for Pb-Sn, see ASM Handbook Vol. 9 |
This issue was explained in section 3.1, lines 268-270 and 281-283. |
28. |
226 The measured XRD (Fig. 6, bottom right) is not fully explained by the presence of lead and hydrocersussite alone. Did you try to match the other peaks?. Common corrosion products of lead would be cerussite and hydrocerussite. In contact to soda glass, NaPb2(CO3)2(OH) was also observed (ICDD-PDF 37-501, see WHEN GLASS AND METAL CORRODE TOGETHER, IV: SODIUM LEAD CARBONATE HYDROXIDE, available at Researchgate). Are any of these phases present? Or what else? |
Thank you for bringing these issues to our attention. We appreciate your feedback, and the given reference is very useful. As such, we have rewritten this part in section 3.1.
|
29. |
230: Figure 7a and b: The signal for SiO2 does not follow the border between lead and glass (as seen in the Pb picture). Why?? |
We have already addressed this issue in the paragraph in section 3.1 (lines 271-278). |
30. |
251: Chanchiaw ?? |
We have already rewritten this section and “Chanchiaw” was deleted. |
31. |
257-259: To explain a difference of glasses in a physical property with the presence of 0.02 % CuO does not make much sense, when the glasses have much higher differences in other compounds (see Table 1) |
We have already addressed this issue in lines 178-182. |
32. |
287: replicated |
We have rewritten the section 3.3, this point has been corrected. |
33. |
300: reproduced |
We have rewritten the section 3.3, this point has been corrected. |
34. |
334/5: …and very likelyly no borax at all! |
We have deleted this paragraph and the issue of borax has been explained din section 2.2. |
35. |
365: What do you mean with ‘isolated CuO atoms‘? |
Actually, we mean Cu0 atoms. This has been corrected in line 479. |
36. |
370: Wrong legend for Fig. 15! (This is legend for Fig. 1) |
We have corrected the legend and changed it to Figure 18 (lines 484-485). |
37. |
373: Wrong legend for Fig. 16! (This is legend for Fig. 1) |
We have corrected the legend and changed it to Figure 19 (lines 487-488). |
38. |
391: Wrong legend for Fig. 17! (This is legend for Fig. 1) |
We have corrected the legend and changed it to Figure 20 (lines 504-505). |

Round 2
Reviewer 3 Report
The authors addressed most of the comments of this reviewer and the quality of the manuscript has improved. This does not change the principle criticism that results are not very useful practically. The not-perfect match of the colours could already be seen with the naked eye, the tightness of the glass-metal bond is irrelevant for conservation purposes. That glass compositions used suffer from weathering could also be found in the technical literature. The reviewer is not convinced that these results are really ‘valuable‘ (l.120), 'crucial' (l. 121)‚ or even ‘useful‘ (l. 533), authors' answer to comment 4 is not persuading.
Of interest are data on the originals (glass and metal composition ). Unfortunately comment 17 and all the questions raised there on lead and its application have not been addressed properly. Were is the table of the metal composition (l. 309)? Table 3 is on the glass composition! As other publications on Kriab do not give alloy composition such a table would give important new information. However, WDS area mappuing or measuring a statictical significant number of individual spots would be needed to obtain the overall composition because of the lateral variation (312ff).
The now reported detection of boron (l. 333) in ancient glass is an extraordinary finding. While borax is mentioned in Venetian recipes since the end of the 17th century, the reviewer is unaware of any detection in glasses made before the end of the 19th century. Can one be sure that the measured samples were really ancient? What date are they? If truely ancient, this would be an important result certainly warranting publication.
However, because the value lies in the measurements on the original ancient (?) samples one can doubt if Materials is a suitable journal. If a re-revised manuscript can supply information on the lead alloy(s) and antiquity of the sample, a journal like Heritage would seem more appropriate.
Further details:
24: Such mirrors are rare. Delete ‘commonly‘
37: artisans?? Better ask professional conservators!
54 (also abstract line 44): While the following reference to India is correct, references for Myanmar and Laos are missing!
60-62: The spread of the technology from India is a hypothesis, not a proven fact. Rephrase:‘…might have spread…‘
92: 3500 BC: reference needed!
95, 497: Glass (per definition) does not have a melting point. Working range for glass blowing ?
104: 900 °C
120-121, 531-533:
Figure 1: Low fotografic quality. Authors should provide better fotografic examples.
146: 900 °C
150: tape?
231: The authors did not explain why they included UV in a weathering test for inorganic materials at all. Would results withot UV (temperature and humidity variation only) be different at all?
318: The intentional addition of Ca and Mg to an alloy in ancient times can certainly be exluded before end of the 19th century!
377 adherend yielding?
408: priceless?
512: analyses (BE)
Author Response
Author's Reply to the Review Report (Reviewer 3)
Note: The line number in the response column relates to the 2nd revised version.
No. |
Comment from referee |
Response |
1. |
The authors addressed most of the comments of this reviewer and the quality of the manuscript has improved. This does not change the principle criticism that results are not very useful practically. The not-perfect match of the colours could already be seen with the naked eye, the tightness of the glass-metal bond is irrelevant for conservation purposes. That glass compositions used suffer from weathering could also be found in the technical literature. The reviewer is not convinced that these results are really ‘valuable‘ (l.120), 'crucial' (l. 121)‚ or even ‘useful‘ (l. 533), authors' answer to comment 4 is not persuading. |
Thank you for taking the time to review our manuscript and providing us with constructive feedback. We appreciate the comments you have provided and have taken them into consideration for the second revision of our paper. We understand that our first prototype Kriab mirrors may not yet match the ancient ones. However, due to the urgent need for restoration work, we have used these prototypes. To avoid any misunderstanding, we have replaced the word 'replicated' with 'alternative' in the second revision of our manuscript. Initially, we attempted to find a way to coat the glass with the resources available to us, knowing that the ancient Kriab mirrors employed lead as a reflective coating. We found that the modified tape casting method, invented by Chanchiaw and his colleagues, was a good starting point, and we used a simple glass composition with high boron at first. In this second revision, instead of focusing solely on the replicated Kriab mirrors, we have also included the results of the chemical composition and mechanical characterization of the alternative Kriab mirrors, as compared to the ancient ones. Additionally, we investigated a variety of ancient Kriab samples in this work, which is a departure from previous studies. We have also removed all the words in lines 120, 121, and 533 from this second revised version.
|
2. |
Of interest are data on the originals (glass and metal composition ). Unfortunately comment 17 and all the questions raised there on lead and its application have not been addressed properly. Were is the table of the metal composition (l. 309)? Table 3 is on the glass composition! As other publications on Kriab do not give alloy composition such a table would give important new information. However, WDS area mappuing or measuring a statictical significant number of individual spots would be needed to obtain the overall composition because of the lateral variation (312ff). |
Thanks a lot for your suggestion. We have already added Table 3 which represents elemental and oxide composition of the reflective coating (metal) layer and glass layer, respectively, of the selected ancient Kriab mirrors, determined by Wavelength Dispersive Spectroscopy (WDS), as in line 341. |
3. |
The now reported detection of boron (l. 333) in ancient glass is an extraordinary finding. While borax is mentioned in Venetian recipes since the end of the 17th century, the reviewer is unaware of any detection in glasses made before the end of the 19th century. Can one be sure that the measured samples were really ancient? What date are they? If truely ancient, this would be an important result certainly warranting publication. |
As the ancient Kriab samples we studied were obtained from Thailand, acquired by specialists from the Office of Traditional Arts, Fine Arts Department, Ministry of Culture, who collected the samples during their restoration work. These provided samples were from antiquities that had never been restored since they were first created by the ancient artisans, making them truly ancient. Most of these antiquities were fabricated in the 18th century, which we have addressed in the new title suggested by the Editor: 'Chemical and Mechanical Characterization of Alternative Kriab-Mirror Tesserae for Restoration of 18th-Century Mosaics in Thailand'. |
4. |
However, because the value lies in the measurements on the original ancient (?) samples one can doubt if Materials is a suitable journal. If a re-revised manuscript can supply information on the lead alloy(s) and antiquity of the sample, a journal like Heritage would seem more appropriate. |
I hope that the corrections made in the second revision of our manuscript will make it suitable for publication in Materials. I also hope that our work will gain high visibility among various fields and readers, and that the findings from our chemical and mechanical results will attract their interest. |
5. |
24: Such mirrors are rare. Delete ‘commonly‘ |
We have corrected this in line 24. |
6. |
37: artisans?? Better ask professional conservators! |
We have corrected this in line 37. |
7. |
60-62: The spread of the technology from India is a hypothesis, not a proven fact. Rephrase:‘…might have spread…‘ |
We have corrected this in line 59. |
8. |
92: 3500 BC: reference needed! |
Thank you for bringing this issue to our attention. I mistakenly wrote 3500 BC when it should have been 3500 years ago. However, we have corrected this point in our manuscript and changed it to 1400 BC. Additionally, we have added references to support this change in line 97. |
9. |
95, 497: Glass (per definition) does not have a melting point. Working range for glass blowing ? |
We have corrected this mistake in both line 100 and line 594. We did not use the glass-blowing technique as used in the Indian method. Instead, we used a modified tape casting method that requires a low viscosity melt. We have explained this technique in detail in section 2.2 of our manuscript. |
10. |
104: 900 °C |
We have corrected this in line 110. |
11. |
231: The authors did not explain why they included UV in a weathering test for inorganic materials at all. Would results withot UV (temperature and humidity variation only) be different at all? |
We have explained this issue in lines 123-124. As these mirrors were frequently used outdoor. |
12. |
318: The intentional addition of Ca and Mg to an alloy in ancient times can certainly be exluded before end of the 19th century! |
Thanks for your comment. Since the Kriab mirrors were fabricated around the 18th century, we found that only one out of three samples in Table 3 contained Ca in the metal alloy layer. |
13. |
377 adherend yielding? |
In the context of adhesive properties, the term 'adherend yielding' traditionally referred to the yield strength of the material being bonded or adhered together. However, to make it clearer and more consistent with common usage, we have revised this term to 'adhered yield strength,' as seen in line 437. |
14. |
408: priceless? |
We have already rewritten the related sentence in lines 472-473. |
15. |
512: analyses (BE)
|
In the second revision of our manuscript, we have decided to use U.S. English style instead of British English, and all necessary changes have been made throughout the text. This makes it easier for us to proofread. |
